# Robust and adjustable dynamic scattering compensation for high-precision deep tissue optogenetics

Zhenghan Li[1,2], Yameng Zheng[3], Xintong Diao[3], Rongrong Li[3], Ning Sun[3], Yongxian Xu[3], Xiaoming Li [3], Shumin Duan[3], Wei Gong [3✉] & Ke Si [1,2,3,4✉]

The development of high-precision optogenetics in deep tissue is limited due to the strong optical scattering induced by biological tissue. Although various wavefront shaping techniques have been developed to compensate the scattering, it is still a challenge to non-invasively characterize the dynamic scattered optical wavefront inside the living tissue. Here, we present a non-invasive scattering compensation system with fast multidither coherent optical adaptive technique (fCOAT), which allows the rapid wavefront correction and stable focusing in dynamic scattering medium. We achieve subcellular-resolution focusing through 500-μm-thickness brain slices, or even three pieces overlapped mouse skulls after just one iteration with a 589 nm CW laser. Further, focusing through dynamic scattering medium such as live rat ear is also successfully achieved. The formed focus can maintain longer than 60 s, which satisfies the requirements of stable optogenetics manipulation. Moreover, the focus size is adjustable from subcellular level to tens of microns to freely match the various manipulation targets. With the specially designed fCOAT system, we successfully achieve single-cellular optogenetic manipulation through the brain tissue, with a stimulation efficiency enhancement up to 300% compared with that of the speckle.

[1] State Key Laboratory of Modern Optical Instrumentation, Department of Psychiatry of the First Affiliated Hospital, Zhejiang University School of Medicine, Hangzhou, China. [2] College of Optical Science and Engineering, Zhejiang University, Hangzhou, China. [3] Liangzhu Laboratory, MOE Frontier Science Center for Brain Science and Brain-machine Integration, State Key Laboratory of Brain-machine Intelligence, Zhejiang University, Hangzhou, China. [4] Intelligent Optics & Photonics Research Center, Jiaxing Research Institute, Zhejiang University, Jiaxing, Zhejiang, China. ✉email: weigong@zju.edu.cn; kesi@zju.edu.cn

Optogenetics has been an important method to understand the functioning of neurons and corresponding connections[1–3] with the help of light and different optogenetic actuators. However, a serious obstacle for the optogenetic manipulation is light scattering and opaque property of biological tissue[4,5]. The light penetration depth, which is commonly represented by scattering mean free path (MFP), is limited. Take visible light as an example, the MFP is usually ten times micrometers in mouse brain tissue[6–8], which means a visible photon will experience tens of scattering events when traveling through 1 mm of biological tissue. Such heterogeneous optical properties (aberration and scattering) among biological samples prevent to obtain a tight focus required for non-invasive, precise optogenetics manipulation inside deep biological tissue.

Although tissue scattering will scramble the incident light field corresponding to the structure of the scattering medium, the scattering does not eliminate the information of the light field, thus the scattering properties can be measured. To decrease tissue scattering, some strategies have been identified using (i) the infrared wavelengths and (ii) wavefront shaping methods. For living tissues, the decorrelation caused by scattering is another challenging due to in vivo activities such as respiration and blood flow during the measurements of biological tissues. Multiphoton excitation relies on infrared wavelengths that are less prone to scattering, provides nonlinear precise generation, and increases the penetration depth[3,9–12]. The dynamic scattering does not affect its performances, however, as depths increase or facing denser tissue, the complexity of the wavefront scrambling can easily overburden the penetration ability of infrared wavelengths. In comparison, wavefront shaping methods have been achieved by the introduction of wavefront modulation devices such as spatial light modulator (SLM), deformable mirror, and digital micromirror devices (DMD), in conjunction with algorithms to figure out the optimal wavfront patterns (regardless of wavelengths) and has the ability to correct severe distortions. Furthermore, the speed and accuracy can be weighed properly to deal with dynamic scattering situations[13–18]. Therefore, wavefront method has long been considered as powerful solution for tissue scattering.

Wavefront shaping strategies counteract the effects of tissue scattering using different properties of the light field (phase, intensity, or propagation direction) to tailor the incident wavefront and modulate the light field, so the incident light can be directed to the target location fast and optimally. Several typical wavefront shaping mechanisms have been reported, including parallel iteration[15,19–26], transmission matrix[22,27–30], and optical phase conjugation[13,18,31–34] or other mainly ultrasound-based methods[17,35,36]. Among these methods, the spatial resolution of ultrasound-based mechanisms is still limited by the ultrasonic guidestar diffraction limit, which is much greater than the optical diffraction limit. The measurement of transmission matrix is time consuming and requires high volume of calculation for higher accuracy. In comparison, the parallel iteration based strategies have the advantages of many different forms, like Hadamard[15,19] or Zernike[19,20] matrix based iteration methods, coherent optical adaptive optics (COAT)[23–25] or the further developed iterative multiphoton adaptive compensation technique (IMPACT)[26], some interferometric methods like focus scanning holographic aberration probing (F-SHARP)[37]. These strategies have been mainly demonstrated in imaging applications with precise corrected focus, while most of them are not demonstrated to optogenetic applications. Besides, deep tissue optogenetics requires that the incident light can be directed to the target position with large possible coverage area, high speed and robustness, improved strategies are expected to satisfy the requirements.

Here, we propose a focusing system using fast multidither coherent optical adaptive optics (fCOAT) mechanism that can form an adjustable focus, from diffraction-limited size to tens of microns. The fCOAT system based on variant strategy is also controllable, stable, high speed, and the binary modulation mode ensures the robustness of the system. Experimental evaluations reveal the focus signal enhancements outperformed much better than the original speckle. We demonstrate the application of this system to deep tissue optogenetics through mouse brain tissues, with significant improvement of the stimulation efficiency and precision. The capability of this approach implemented neural signaling studies is also revealed.

## Results

**System design and operating principles**. To focus through scattering media achieving precise enhancement of stimulation, we designed a fCOAT system which contains a focusing part for stimulation, an imaging part for the monitoring of calcium signal, and the formed focus (Fig. 1, Supplementary Method 1, Supplementary Fig. 1). The scattering media and the biological samples were placed between the two objective lenses. In order to facilitate observation of cultured neurons in the system, the imaging part were designed as inverted so the scattering media can be easily placed at the top cover of the petri dishes. There is no need to open the cover for imaging which ensures the neuron viability. The focusing part used low numerical aperture (NA) lenses to ensure the working distance is long enough to place the scattering media and petri dishes. While the imaging lenses vary according to different samples, 20× and 40× objective lenses were mostly used. For the neuron activation, a shutter was used to switch the focusing light on and off at a proper frequency that matches the requirements of optogenetic actuators.

In order to generate a focus through dynamic scattering medium, we used an algorithm modified from parallel iteration optimization method that operates with high speed and robustness with the use of a ferroelectric liquid crystal based SLM (FLC-SLM) which has a net latency of ~ 1 ms. In our system, the FLC-SLM is conjugated to the rear pupil plane of the focusing objective. The SLM pixels are segmented into $N \times N$. The total number of measurements used as feedback is related to the N value, and is less than $4N^2$ with our design. Different N values were taken according to the requirements of speed and accuracy. A compensation profile was generated after one iteration of the modulation process was completed, and the number of iterations can be increased to improve the accuracy. To maintain the high modulation speed, we did this process once to increase the speed in dynamic scattering situation. Details about the system working process are provided in Methods and Supplementary Fig. 2, 3.

**The performance of fCOAT focusing system with thick biological tissue**. The performance of fCOAT system was experimentally verified by measuring the directly transmitted light field scrambled by highly scattering brain slices and/or mouse skulls. The laser was tuned to a power of ~500 μW after the focusing objective to ensure the feedback signal is sufficient enough and attenuates before the camera to prevent overexposure. The scattering medium was placed at the distance of the height of petri dishes away from the focal plane of the imaging lens, which is same as in biological experiments (Fig. 2a). The transmitted light field intensity profiles were both acquired at the start of the correction process and the end of it.

The original scattering speckle images and the corrected focus images through brain slices of different thicknesses are shown with intensity profiles through the center of the corrected region along the dashed line (Fig. 2b, c). Striking differences can be seen after the modulation completed, where the fCOAT system achieves orders

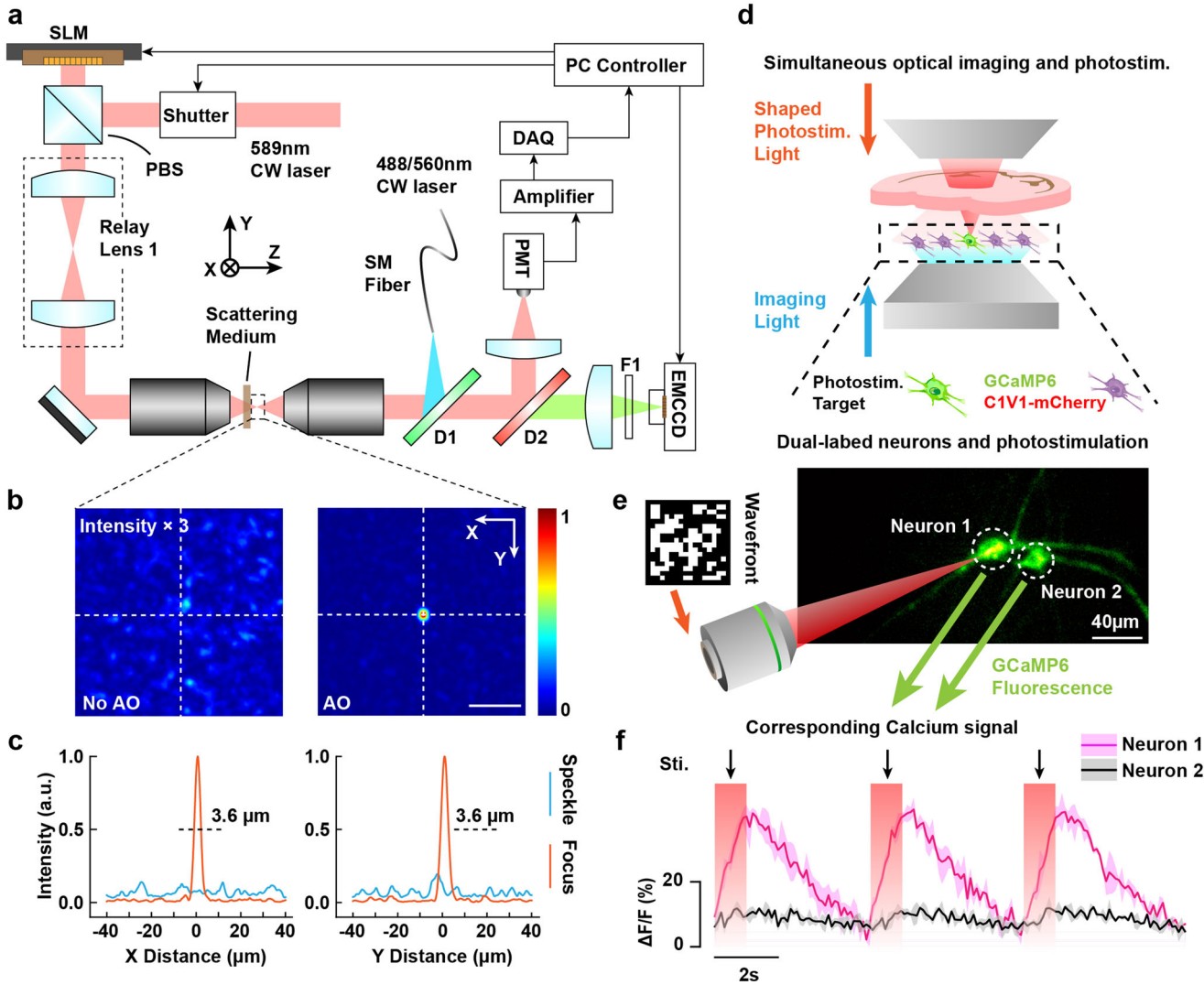

**Fig. 1 fCOAT focusing and corresponding optogenetic manipulation schematics. a** fCOAT focusing system using feedback signal collected from PMT. **b** Speckle and reformed focus pattern using fCOAT system. Scale bar, 20 μm. **c** Intensity section profiles marked in **b**. **d** Optogenetic manipulation schematics using fCOAT system. Shaped photostimulation light formed a tight focus after passing through biological tissue. Neurons expressing both the calcium sensor and optogenetic probe were selected as targets for stimulation. **e** The recorded images of target neurons with shaped photostimulation light illumination. Scale bar, 40 μm. **f** Calcium signals acquired from images recorded as shown in **e**. Shadows are given by the upper and lower limits of stimulations.

of magnitudes signal enhancement higher than original unmodulated speckles. The signal enhancement is represented by the peak-to-background ratio (PBR), which reaches about 200 at maximum. In addition, the sizes of the focus were measured, the full width at half maximum (FWHM) reaches about 3.6 μm, which is its minimum and typical value that matched the optical diffraction limit, when the scattering is less severe (100 μm brain slices). After the modulation completed, the fCOAT system formed a bright and well-defined focus that can be used for imaging or optogenetics manipulation. From 100 to 500 μm brain slices, the focusing performances are continuously reducing (Fig. 2d, Supplementary Method 2) due to the limited correction ability that depends on the modulation segments and the gradually increasing scattering and absorption of the brain slices. However, the FWHM is still controlled to be less than 4 μm. From the acquired images, the actual position of the focus and the pre-defined focus position were compared. The position of the actual focus here is defined as the peak intensity point (Fig. 2e). Using the intensity of the scattered light of the pre-defined focus point as the feedback signal, the

fCOAT system keeps the focus position stable and in control after the modulation, however, the speckle pattern does not determine the location of the main light intensity distribution. Note that the magnitudes of the insect line profiles are normalized to the maximum fCOAT system corrected signal.

The performance of the fCOAT system with severe scattering is further investigated by using different pieces of intact mouse skulls as the scattering medium (Fig. 2f). The scattering speckle and corrected focus images are also shown with intensity profiles through the center of the corrected region along the dashed line (Fig. 2g). Higher than one order of magnitude PBR enhancement is obtained and the maximum average value reaches about 45 after through one piece of skull. Compared to that of using brain slices as the scattering medium, the enhancements are lower when the thickness is similar (Fig. 2h). This is because the skulls are denser than the brain slices and leads to severe scattering. However, a clear and well-shaped focus with considerable PBR is generated after the modulation when facing three pieces of overlapped skulls.

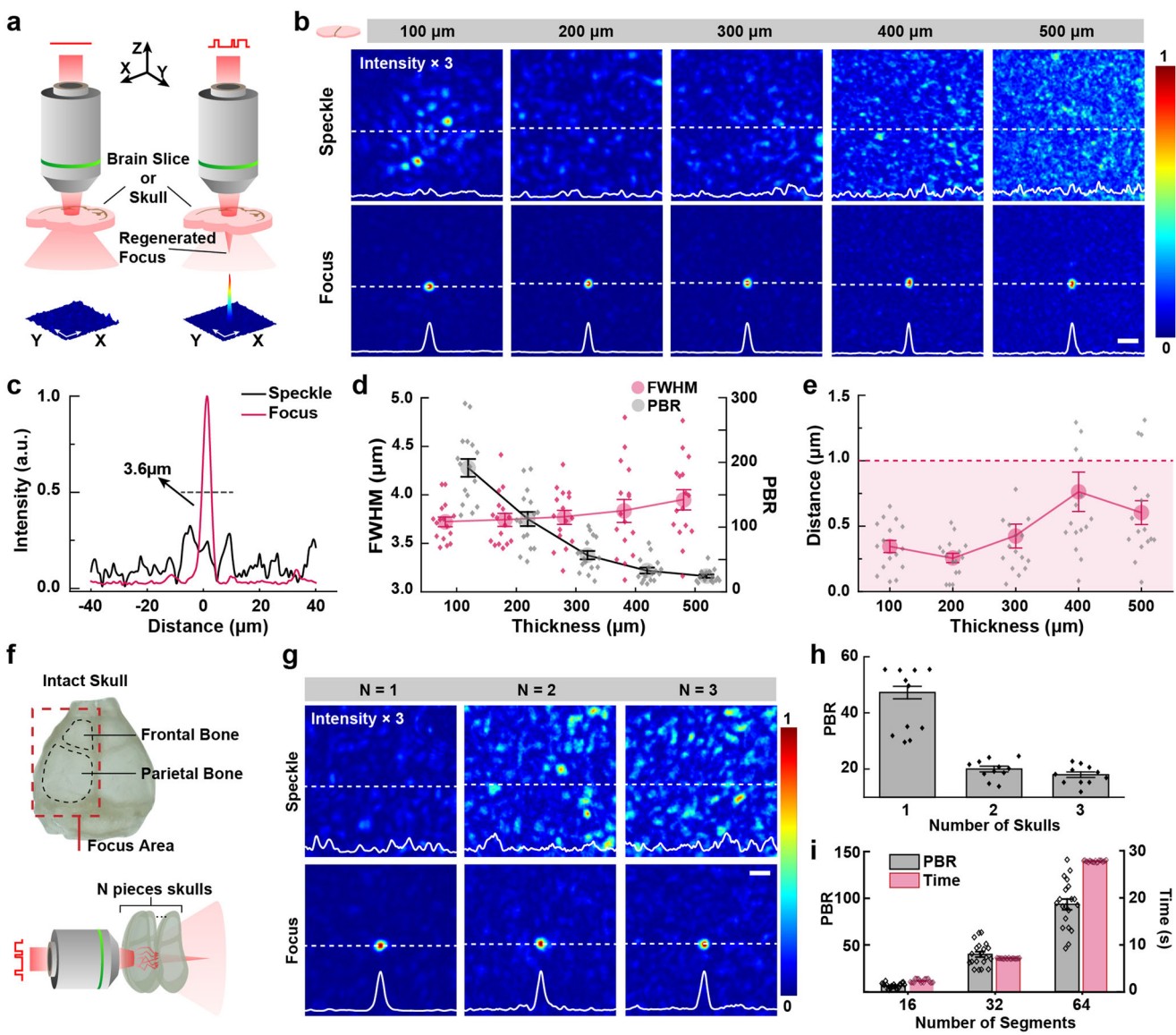

**Fig. 2 A comparison of fCOAT focusing and conventional focusing through biological tissue. a** The operating modes using fCOAT system and conventional focusing system. A tube lens and a camera used together with the imaging objective are not shown. **b** Images of the conventional and fCOAT generated focus profile through brain tissue slices (100, 200, 300, 400, and 500 μm thick). The intensity profile of the dotted section line in the figures are shown at the bottom edge of the images. Scale bar, 20 μm. **c** An intensity profile selected from the data through 200 μm brain slices. **d** The statistical data of the full width at half maximum (FWHM) focal spot sizes and the peak to background ratio (PBR) of the focus as the functions of tissue thickness. **e** The statistical data of the distance between focus position and the expected position as a function of tissue thickness. The shadow in **e** indicates the range below 1μm. Error bars in **d**, **e** represent the SE of sixteen measurements taken at random different locations on brain slices. **f** Schematic of a mouse skull. **g** Images of the conventional and fCOAT generated focus profile through different layers of mouse skulls. The intensity profile of the dotted section line in the figures are shown at the bottom edge of the images. Scale bar, 20 μm. **h** The statistical data of the PBR of the focus as the functions of layers of mouse skulls. **i** The statistical data of the PBR of the focus and corresponding time consumption as functions of number of used SLM segments. Error bars in **h** and **i** represent the SE of eleven and twenty measurements taken at different locations within the red dotted frame of the mouse skull in **f** respectively.

The trade-off between modulation speed and accuracy is investigated by completing the modulation using different SLM segments to find optimal parameters when facing dynamic scattering medium. The FLC-SLM used in the system have $512 \times 512$ pixels, and we chose three different segmentations, $16 \times 16$, $32 \times 32$, and $64 \times 64$, respectively. Using one piece of intact mouse skull as scattering medium, we recorded the corresponding scattering, corrected images and time consumption of the modulation. The PBR of the generated focus and the time consumption were calculated and compared (Fig. 2i). Considering the unstable performance ($16 \times 16$ segments) and excessive time

consumption ($64 \times 64$ segments), using $32 \times 32$ segments is still the optimal solution for our system which has both considerable signal enhancements and relatively less time consumption (~7 s).

The signal enhancement of fCOAT system is relatively limited due to the binary-mode modulation, however, more SLM segments were used to compensate for the lower modulation depth considering the high modulation speed. With this expectation, the performance of the system is verified experimentally with brain slices and intact skulls and the optimal parameter for the system is also investigated. The working process is also modified to increase the speed and accuracy as discussed in Methods in detail.

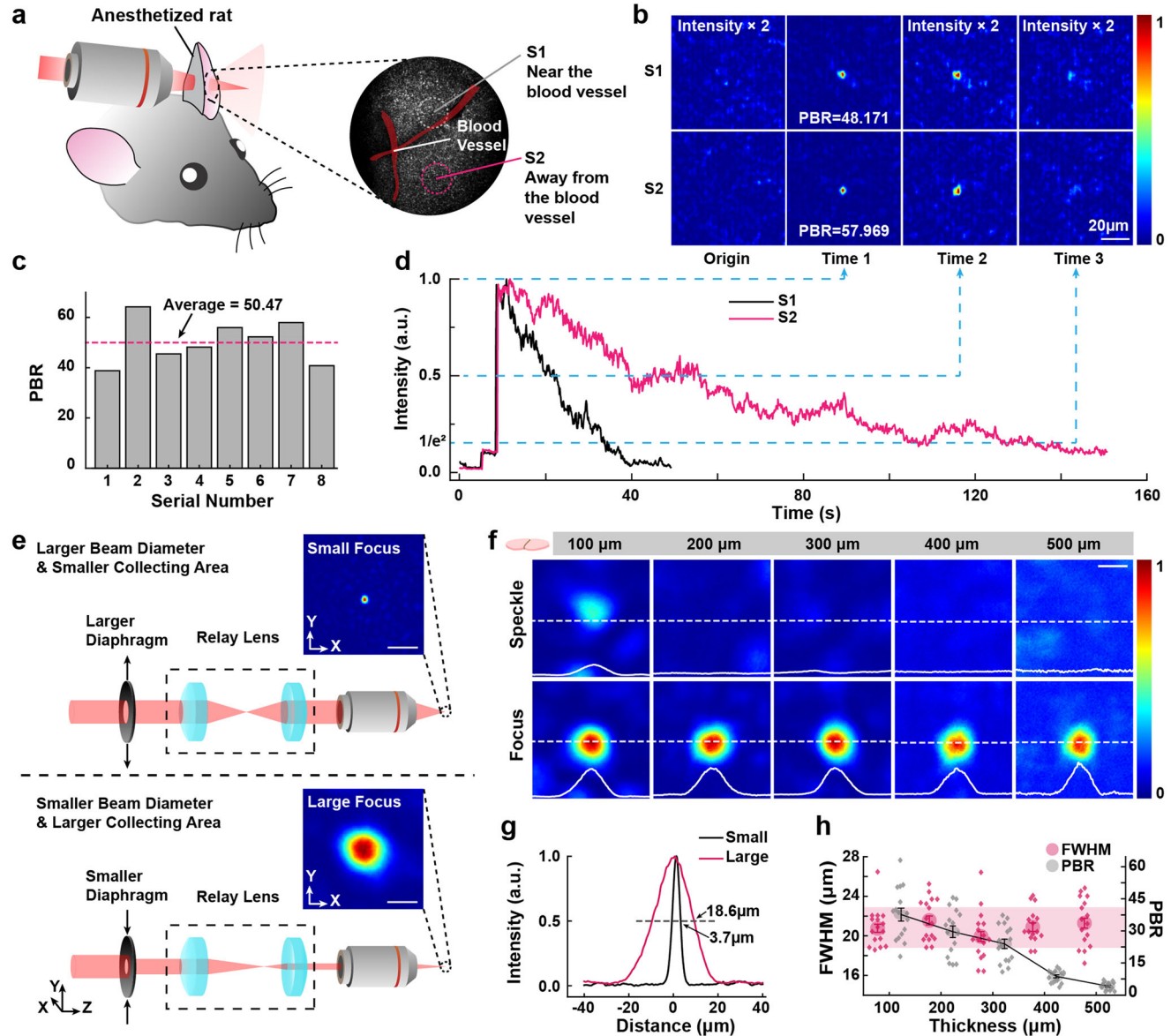

**Fig. 3 The focusing ability under different applications. a** Handling of the experimental mouse and the schematic of the focusing position. **b** The focus PBR right after the focusing procedure finished at several positions on the mouse ear. **c** Images of the conventional and fCOAT generated focus profile through living mouse ear at two different positions shown in **a** at different time points. Scale bar, 20 μm. **d** The intensity profiles of the peak position of the focus shown in **c** as a function of time. **e** Schematics of the generation of small (top) and large (bottom) focus. Scale bar, 20 μm. **f** Images of the conventional and fCOAT generated focus profile through brain tissue slices (100, 200, 300, 400, and 500 μm thick). The intensity profile of the dotted section line in the figures are shown at the bottom edge of the images. Scale bar, 20 μm. **g** A comparison of the intensity profiles between small and large focus through 200 μm brain slices. **h** The statistical data of the FWHM focal spot sizes and the PBR of the focus as the functions of tissue thickness. The shadow represents the centralized area of FWHM data points. Error bars represent the SE of sixteen measurements taken at different locations.

**Adjusted focusing mode for biological studies**. Regarding the decorrelations of scattering in living tissue, the performance of fCOAT system against dynamic scattering would have a great impact for optogenetics manipulation. We used a live rat ear as the dynamic scattering medium to verify the fast-focusing ability and robustness of our system. The rat ear was measured to be about 400 μm. The SLM was separated to 1024 (32 × 32) segments. The laser was tuned to a power of ~ 500 μW after the focusing objective. The focusing process was carried out at different positions of the rat ear several times (Fig. 3a). The focusing was monitored and acquired at 10 frames per second from the beginning to the end of the focusing procedure.

The scattering speckle images and the corrected focus images of different positions of the rat ear are shown, along with the images at the other two different time spots after the modulation completed (Fig. 3b). Due to the decorrelation of the scattering pattern, the average level of the signal enhancement is lower than the results acquired from static brain slices of similar thickness (Fig. 3c), while obvious differences are observed after the modulation completed. The fCOAT system achieves about 50 times signal enhancement and the focus sizes remains at optical diffraction limit level (FWHM~3.6 μm). The pulse and blood flow of the rat ear disturb the incident light and lead to the signal decrease. The intensity profile of the focus position is plotted (Fig. 3d). The position near

the blood vessels shows faster signal decreasing than the position further from the blood vessels because of its strong disturbing of the blood vessels. The experimental results show that the focus last longer than 40 s when the target positions are further from the blood vessels, even at the positions near the blood vessels where the signal decorrelates faster, the focus can last longer than 10 s. Because of the binary modulation strategy, the robustness of the fCOAT system is higher than the full phase modulation and the focus have the ability to maintain longer during the decorrelation of the living tissue.

To further meet the requirements of the optogenetics manipulation, the focus size is investigated to match the target neuron size. The neuron size is mostly greater than the diffraction-limited focus, so the focus size was designed to be adjustable to cover the target better (Fig. 3e). We used the same laser power and verified the ability of our system to obtain large focus experimentally. We collected the original scattering images and the corrected focus images through brain slices from 100 to 500 μm with intensity profiles through the center of the corrected region along the dashed line (Fig. 3f). We expanded the focus size using lower NA (~1/5 of the pervious) of the focusing module (Fig. 3g). The expanded focus FWHM were measured to be about 20 μm, which is in line with our expectation. The focusing performance is also decreasing with the increasing scattering and absorption of the brain slices, while the focus is still clearly recognizable (Fig. 3h) with the maximum average PBR of the value about 35. The details about the system adjustments for the larger focus are discussed in Supplementary Method 3.

The robustness of fCOAT system mainly arises from the high modulation speed and the higher tolerance of phase errors that caused by the changes of scattering medium, because the binary modulation mode is an averaging of full phase modulation and thus allows greater perturbations. The ability of the formed focus through live rat ear is a typical example of these two features.

The robustness can also be reflected through the ability to obtain a large focus. The expanded focus can be regarded as a combination of several small focus, the intersection of the phase pattern correspond to each focus is the majority of the final phase pattern to generate a large focus. The binary phase modulation has lower PBR than full phase modulation, but the binary pattern leads to smaller conflicts and the gap with PBR of full phase modulation will not be further widened. Moreover, the signal enhancement maintains at higher than an order of magnitude although lower than small focus.

**The manipulation of neural activities using the fCOAT system.** With all the demonstrated focusing performance of fCOAT system, we used it to implement precise optogenetics enhancements. A 200 μm thick brain slice was used as the scattering medium. The cultured hippocampus neurons at 8–10 days in vitro were co-transfected with optogenetic actuators (C1V1) and calcium indicators (jGCaMP6s), and the regenerated focus was located inside the target neurons and outside them (speckle), respectively (Fig. 4a, b). The corresponding calcium signals were acquired as image sequences at 10 frames per second using wide-field fluorescence imaging. The 589 nm laser power was set to ~ 500 μW after the focusing objective, which is enough for light stimulation. The 488 nm laser power was <12 μW/mm$^2$ to avoid C1V1 being activated by 488 nm light[38].

The co-localization of the regenerated focus and target neurons were monitored using the same camera but different switchable filters. The position of the focus is determined by the position of photomultiplier tube (PMT) and would not be influenced by the scattering medium. The focus position was localized first and the neurons easily moved to overlap the focus. The co-localization was confirmed before each experiment to ensure the accuracy.

We gave 200 ms of stimulation (stimulation period of 5 s) using focus and speckle patterns, respectively, the calcium signal of stimulations (before and after) is shown in Fig. 4c. The activation effect is reflected by the cellular calcium signal intensity, which is characterized by ΔF/F, calculated according to the reported method[39]. The calcium response follows the stimulation very well, and the focus stimulation lead to a higher ΔF/F than that of the speckle stimulation (Fig. 4d, e). The mean value of ΔF/F becomes 70% higher when the neuron is stimulated by focus, while the value is only 22% using speckle stimulation. The performance of the system is further verified by implementing the experiments on different hippocampus neurons. The stimulation time and the period were adjusted according to the neuron state, observing surprising responses enhancements (Fig. 4f, g, Supplementary Fig. 4). Results from different neurons show different enhancements, which is the inevitable result of the combination of two factors-focus parameters and neurons themselves. The focus parameters-PBR, sizes and the enhancement of the calcium responses show a positive correlation (Fig. 4h). Meanwhile, the expression of the optogenetics actuators in neurons also influences the experimental effects. We statistically analyzed the co-expressed images of the neurons (Fig. S4) and found that the well expressed neurons exhibit stronger enhancements of calcium responses after stimulation, while those neurons with poor expression show less enhancements.

The fCOAT system was designed to be adjustable and optimized according to the characteristics of biological samples. In this experiment, we selected Channelrhodopsin-2 (ChR2) as the actuator and Cal590 as the calcium indicator to reduce interfere between activating and imaging light. Under this condition, we set activating laser wavelength at 488 nm and the imaging laser wavelength at 560 nm. The 488 nm laser power was adjusted to ~ 300 μW after the focusing objective, 560 nm laser power was tuned to <1 mW/mm$^2$. In addition, we expanded the focus size to adapt the size of target neurons (Fig. 4i, Supplementary Fig. 5a). The power adjusted here is about the maximum value that does not evoke neural activity with speckle stimulation, while the neurons remain responsive using focus stimulation under the same light intensity. The intensity profiles extracted from the fluorescence images verify that using suitable stimulation intensity and larger cover area enables more precise activation (Fig. 4j). The above experimental results show that the system can be easily adjusted to achieve optimal experiment conditions, where the modulated light can accurately stimulate the target neuron without affecting the surrounding neurons.

**Neural signaling research using fCOAT focus stimulation.** To explore potential optogenetical applications, the neural signaling was studied using our fCOAT system. Neural connections are suggested to play an important role in brain information storing and retrieving[40–44]. However, some small stimulus changes can lead to different degrees of responses or follow-up results for neurons[45]. Therefore, we believe that whether neurons can be stimulated effectively and relatively precisely would have different effects on neural populations.

Here, we performed two kinds of stimulations (large focus and speckle) at the same frequency on rat hippocampal neurons cultured for 12 days in vitro (ChR2 & Cal590), analyzed the synchrony of experimental results and reconstructed underlying clustering and intrinsic connections in neurons (Fig. 5). For the same neuron population, we gave 1 s of stimulation (stimulation period of 5 s, the stimulation power was set to ~500 μW before scattered by brain slices), that is, 9 corresponding stimulations within 45 s (Supplementary Fig. 5b). Neural segmentation was achieved using the collected data to extract calcium response

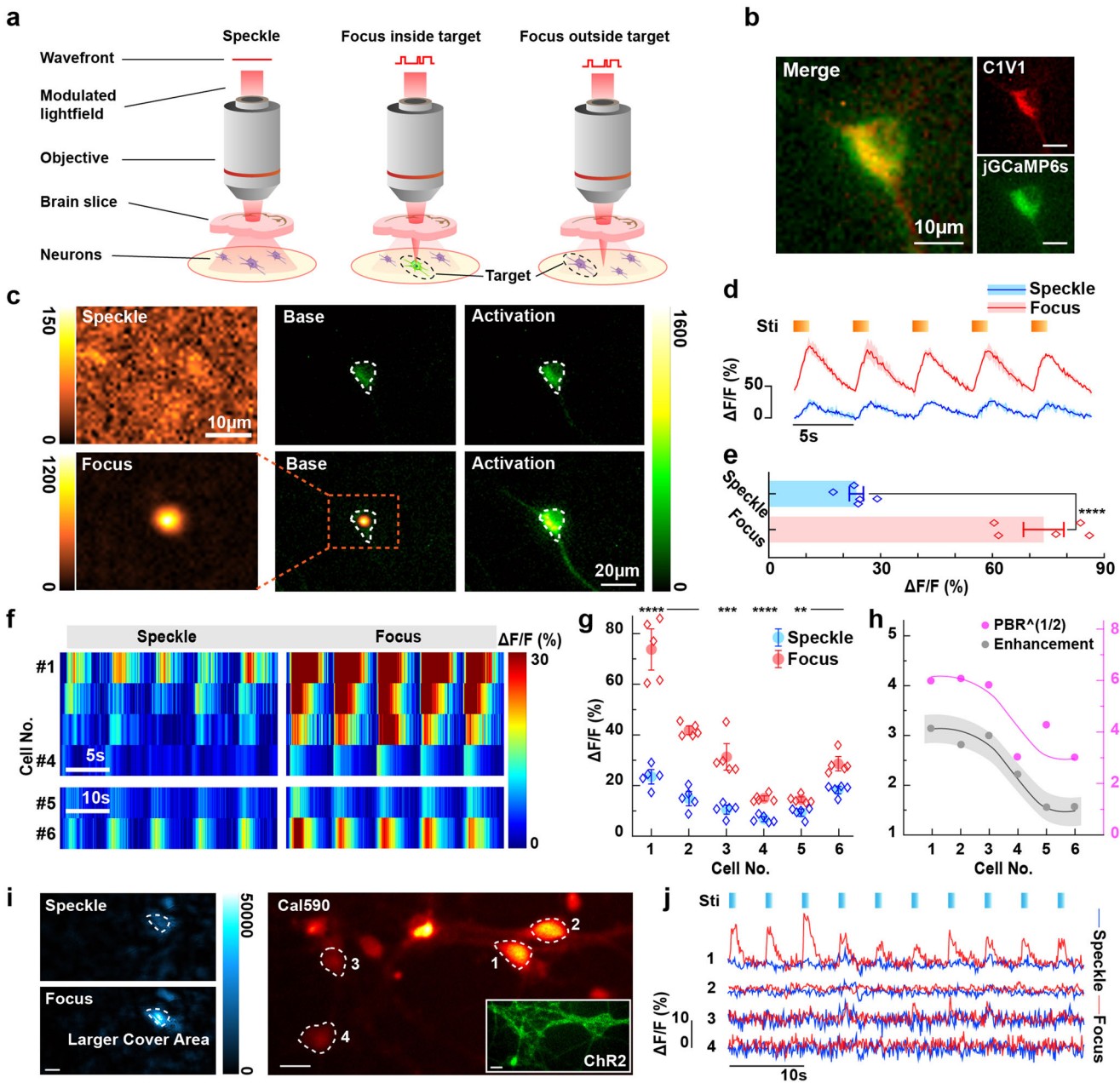

**Fig. 4 The manipulation of fCOAT system on neural activities. a** Diagram illustrating the experimental scheme used to demonstrate the ability of fCOAT focusing to elicit stronger calcium signals through 200-μm-thick mouse brain tissue. The black dashed line indicates the positional relationship between the target neuron and the focus. **b** The merge images of the expression of photosensitive protein and calcium indicator of the target neuron. Scale bar, 10 μm. **c** Targeted activation of calcium signaling in a representative neuron after focusing through the brain slice. Green: jGCaMP6s fluorescence signals. Orange: excitation beam intensity through the brain slice with and without wavefront shaping. The white dotted line: the location of the optimized focus and the boundaries of individual neuron. **d** Quantitative analysis of jGCaMP6s fluorescence signals obtained from the neuron in **c**. Shadows are given by the upper and lower limits of stimulations. **e** The histogram representation from the data in **d**. Comparison of peak neural response (mean ± SE) to different types of stimuli, ****$P < 0.0001$, strong (73.76 ± 5.41% ΔF/F) versus weak (23.47 ± 1.89% ΔF/F). **f** Individual traces of jGCaMP6s fluorescence in different C1V1 neurons under 589 nm illumination using focus and speckle light fields. **g** The statistical results of neuron data in **f** and the corresponding analysis between different types of stimuli. Error bars:1.5 SE of five measurements of peak neural responses. **h** The results of the relationship between the focus PBR and the calcium signals enhancements. The shadow is formed by moving the trend line up and down. **i** Targeted activation of calcium signaling in representative neurons after focusing through the brain slices. Red: Cal590 fluorescence signals. Green: the expression of the used opsin ChR2. Blue: excitation beam intensity through the brain slice with or without wavefront shaping. The white dotted line: the boundaries of individual neurons. Scale bar, 20 μm. **j** Individual traces of Cal590 fluorescence in different ChR2 neurons under 488 nm illumination using extended focus and speckle light fields.

(Supplementary Fig. 5c). We ruled out the possibility of multi-point stimulations and considered that the two different stimulation modes can be regarded as a single influencing factor for this neural population. Taken the stimulation target and the surrounding neurons (#19, #20, #21) as examples, neuron #19 does not express ChR2, while the other two both expressed, and our stimulation target neuron is #21. With speckle stimulation, neither #19 nor #20 produce effective follower responses at the

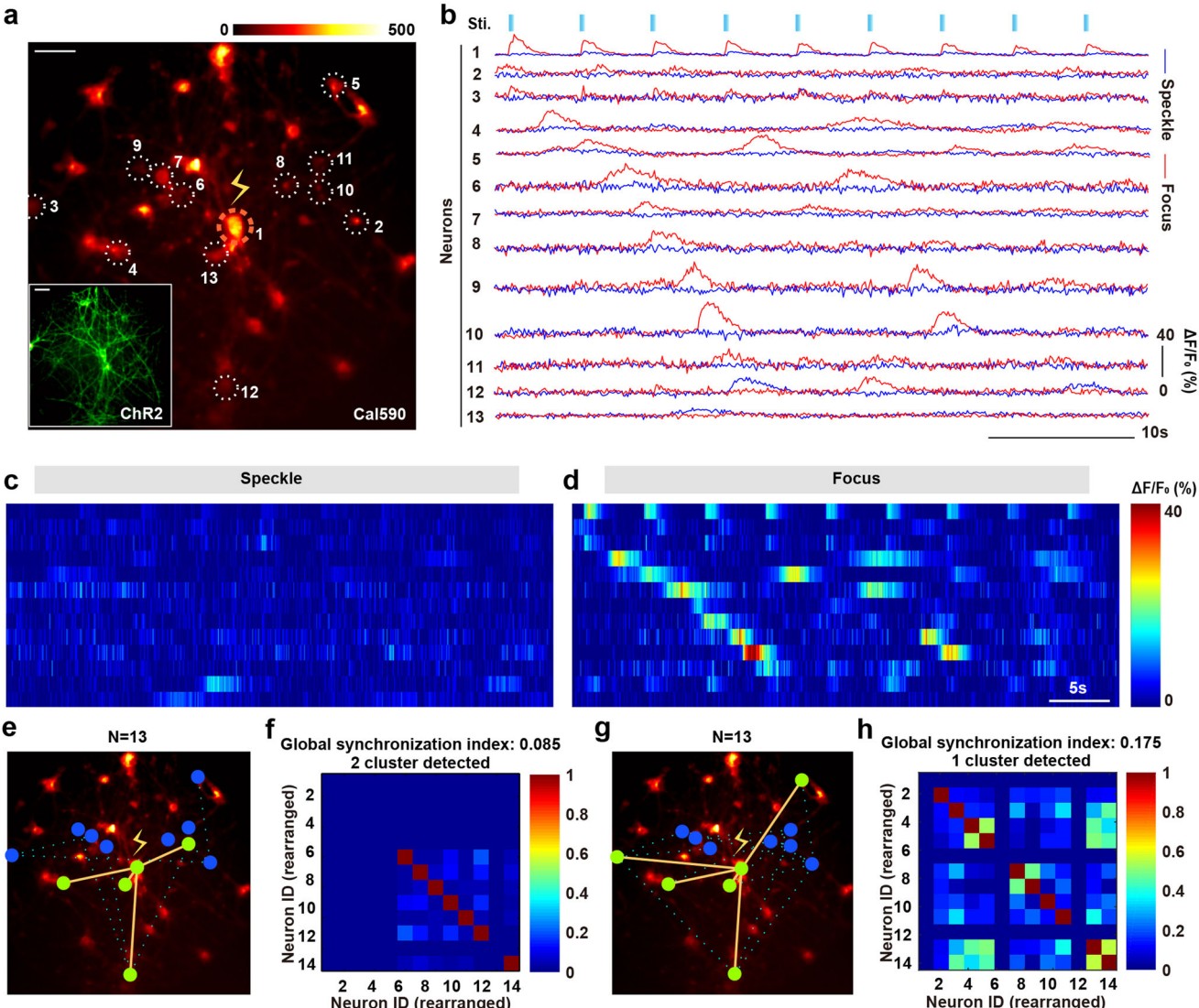

**Fig. 5 Neural signaling research with the use of fCOAT generated focus stimulation. a** Targeted activation of calcium signaling in representative neurons. Red: Cal590 fluorescence signal. Green: the expression of the opsin ChR2 used. Neuron 1 is the target stimulation neuron, and the rest are neurons that follow the response. **b** Individual traces of Cal590 fluorescence in 13 different ChR2 neurons under 488 nm illumination, using extended focusing and speckle light fields. The red line: focus, the blue line: speckle. **c** Individual traces of Cal590 fluorescence in different ChR2 neurons under 488 nm illumination using speckle light fields. **d** Individual traces of Cal590 fluorescence in different ChR2 neurons under 488 nm illumination using focus light fields. **e** Speckle light activation of neuron 1 triggers estimation of functional connectivity of different neurons. Functional connectivity of pairs of neurons (orange line, green dots) centered on neuron 1 is determined by testing the statistical dependence of neuronal calcium activity. **f** Quantification of Synchronization Index and cluster number under speckle light illumination. **g** Focus light activation of neuron 1 triggers estimation of functional connectivity of different neurons. **h** Quantification of Synchronization Index and cluster number under focus light illumination.

time of stimulation (Supplementary Fig. 5b, c), which is the same with focus stimulation, while the calcium responses of neuron #21 are weaker with speckle stimulation than that of using focus stimulation. Therefore, the results indicate that only the intensity of the focus point can be improved to the activation threshold, which is the only worthy difference between the two stimulation modes, in line with our point of view.

After the neural segmentation, we obtained waveforms of 34 neurons, and finally selected 13 neurons which produced at least one effective calcium response (Fig. 5a). Comparing two groups of calcium response curves with different neurons, we found that the focus stimulation leads to better responses of target neurons (#1), characterized by higher ΔF/F, the following calcium responses of the nearby neurons are also much more obvious than the speckle stimulation because of highly efficient stimulation effect (Fig. 5b,

Supplementary Fig. 5d, e). Then, we differentiated subgroups of co-synchronized bursts under the two stimuli by analyzing the corresponding correlation matrices. In the synchronization analysis mode, both stimulation modes are employed to the same group of neurons, and the focus mode has a higher synchronization than the speckle mode (Fig. 5c, d), which indicates that the target neuron (#1) is better activated with focus stimulation and leads to obvious neural transmission. According to the reconstruction of the neuron connection, with the neuron #1 of the light stimulation point as the center, the speckle stimulation method induces the synchronous response of neurons (#4, #10, #12 and #13) (Fig. 5e). However, the focus method produces slightly different results, which means its central evoked neurons (#1, #4, #12, #13) is basically the same, but different in long-distance transmission (#3, #5) (Fig. 5g). At the same time, we used

FluoroSNNAP to analyze the calcium signal synchronization index of neurons under two kinds of light stimulation, and found that the synchronization of focus is significantly better than that of the speckle (Fig. 5f, h). In addition, it is found that the number of neuron clusters in focus mode is one and the number in speckle mode is two (Fig. 5f, h).

The greater the number of neuron clusters suggests that the presence of two or more different neurons as the dominant triggering neural responses. Since only the middle neuron is stimulated, the number of neuron cluster should be one. The result using the speckle stimulation shows that the number of neuron cluster is two, indicating that other neurons are affected by something other than the target neuron. Calcium responses of neurons are evoked more precisely using focus stimulation, and the responses of neurons follow the stimulation much better. While the stimulation efficiency is not enough to produce clearly visible calcium response fluctuations with speckle stimulation, and no further neural transmission analysis can be implemented. We believe that our system is promising in such single-neuron-accuracy optogenetical applications.

## Discussion

Compared with recently developed strategies which mainly focus on basis selection, the fCOAT strategy is suitable for high spatial frequency wavefront distortion due to random scattering, and has relatively high speed and robustness. Several strategies based on Zernike basis have been applied to multiphoton imaging, where low spatial frequency aberration has occupied major proportion. In contrast, the fCOAT strategy measures and compensates for both aberration and high spatial frequency distortion, which is demonstrated experimentally in Figs. 2 and 3. As a result, the fCOAT strategy has higher focusing efficiency than the Zernike based strategies since the randomly scattered light can be coherently delivered to the focus position.

Higher frequency matrices like Hadamard or totally random matrices are also used as basis in iterative strategies, which compensate for high spatial frequency distortion like the fCOAT strategy. For these strategies, the efficient modulation bases are limited to a little part of the whole used bases, and the robustness in dynamic living tissues is not demonstrated. The total measurements to complete the compensation for the scattering can be enormous. While the number of the measurements of the fCOAT strategy is steady and small, less than 3500 when the number of SLM segments is 1024, which ensures high modulation speed that can be increased further with the help of faster modulation or electronic devices[46]. And the robustness is also demonstrated experimentally in vivo. Moreover, continuous working mode is also simulated after a complete wavefront compensation by adjusting initialization parameters (Supplementary Fig. 6a, b). This ability is beneficial for compensating for drift in focus and maintain it for a longer period of time[15,28].

Different from other strategies, we experimentally demonstrate the ability of the system to form a larger focus, which is an intersection of diffraction-limited focus. There are also some strategies have mentioned such multi-focus-like results. However, they both request acquiring a large area of speckle images as the feedback, which both increased the time in acquiring and the computational burden. Here, we only collect the single-dimensional feedback signal from PMT, using a pinhole or a MMF to limit the collecting area. Since the collecting plane is conjugated to the imaging/target plane, we could adjust the size, shape or position of the pinhole to control the resulting focus status (Supplementary Fig. 6c, d, e), and also only a single-dimensional signal is required as feedback. In contrast, acquiring images as feedback can be time-consuming.

Living biological tissues are mostly heterogeneous and dynamic, which are major obstacles for high-precision deep tissue optogenetics. We experimentally demonstrate the ability of the system to focus light through brain tissues of various thickness and various pieces of skulls. Considering that craniotomy or thinning the skull down to a thinner thickness is required even in multiphoton based optogenetics manipulation, our results with skulls and dynamic tissues demonstrate extended penetration depth in living tissue, and the potential to focus light inside living bodies with least damage, which benefit to noninvasive deep tissue optogenetics. Using the wavelengths mentioned above, our system increases the penetration depth up to 500 μm. Considering the penetration depth of long wavelength (infrared) light is much longer than the visible light, the system can increase its working distance to a brand-new level. Furthermore, using long wavelength multiphoton-excited fluorescence as the feedback signal[26], a reflection setup can be implemented to noninvasive in vivo experiments.

Moreover, the large coverage area of the enlarged focus enables more efficient stimulation. Patterned stimulation and scanning stimulation are both used for large coverage area in multiphoton mode, which are based on the fact that the infrared wavelengths are less prone to scattering and not demonstrated in severe scattering scenarios. Our system provides an approach to solve both the scattering and the coverage area requirements, which benefits deep tissue optogenetics and results of precise optogenetics stimulation enhancements are experimentally demonstrated as shown in Fig.4.

It is known that spontaneous reactions may have influences in neural connection reconstructions, which we do not exclude for two reasons. First, when a neuron is effectively stimulated, some possible spontaneous reactions can be masked. Second, in real application scenarios, the spontaneous neurons will not be inhibited, especially in vivo. We have demonstrated that the focus stimulation is far superior to speckle stimulation experimentally, both in focus precision and stimulation effectiveness, thus the application of this strategy to neural network exploration or in vivo behavior regulation can be implemented in the future.

We report a fCOAT focusing system that has super penetration depth and benefits for the non-invasive deep tissue optogenetics. The system demonstrates an ability to form a focus through highly scattering mouse brain tissue, mouse skulls, and dynamic live rat ear. The results show that the focus formed with a single fCOAT iteration are well-defined with high PBR, and remain valid ~60 s with live rat ear. Meanwhile, adjustable focus size is offered to match the requirements of coverage area. The system achieves significant enhancements of stimulation responses for target neurons. The interference of nearby neurons is reduced due to the fCOAT system suppresses random scattering and delivers power more efficiently to the focus. The advantages of adjustable focus, high speed and robustness, high penetration depth make the fCOAT system become a promising tool for noninvasive deep tissue optogenetics manipulation. It also paves the way for precise stimulations at unprecedented depth that is benefit to in vivo neural studies involving highly turbid samples including, but not limited to, neuroscience and brain science.

## Methods

All mouse related experiments were carried out following protocols approved by the Institutional Animal Care and Use Committees of Zhejiang university of China. All mice used in this study are group housed with a 12/12-h light/dark cycle (lights on at 07:00), controlled temperature (25 °C) and humidity (60%) and free access to food and water.

**Brain slice preparation**. All used brain slices (100 to 500 μm) were from 8- to 10-week-old C57BL/6 J mice. Mice were deeply anesthetized with 1% sodium pento-barbital solution, followed by the transcardial perfusion with 40 ml of ice-cold Phosphate Buffered Solution (PBS) and 20 ml of ice-cold 4% PFA. All solutions

were perfused at a uniform rate of 10 mL/min. Mouse brains were harvested and immediately placed in 20 ml of ice-cold 4% PFA and incubated at 4 °C for 24 h to allow penetration of fixatives. The brain was then sectioned with a vibroslicer (no. VT1200S, Leica) into 100 to 500 μm slices. To mount brain slices, stainless steel square holder (25 × 25 mm with a 10 × 8 mm hole) with appropriate thickness was glued to a glass slide (25 × 50 mm). The holder was filled with PBS and glycerin (1:1 mixed) and the brain slice was placed into the holder. Another glass coverslip (23 × 23 mm) was covered on the holder to block the tissue from the air.

**Rat ear and mouse skull preparation**. For dynamic scattering media, a live Sprague-Dawley (SD) rat ear was used to maintain the expected life activities that induced fast decorrelation of the scattering pattern. A 2-month-old male SD rat was used and injected 3% (w/v) sodium pentobarbital into the abdominal cavity at a dose of 1 ml/kg to anesthetize the rat. We used hair removal cream to clean the hair on its ears. Then we used two transparent acrylic plates with holes to clamp one of its ears.

The used whole dissected mouse skulls were from 8- to 10- week-old C57BL/6 J mice that were anesthetized with isoflurane (4% for induction and 1–2% during surgery). To minimize inflammation and brain edema, we injected dexamethasone (2 mg/kg) subcutaneously prior to surgical incision and applied ointment to the mouse's eyes to maintain their moisture. We removed the skin atop the cranium and cleaned the skull surface with a scalpel while continuously perfusing saline over the surgical area. Next, the entire skull edge was thinned using a 0.5-mm-diameter drill, until skull was disconnected from the surrounding tissue. Then, we perfused saline throughout the drilling. Finally, we removed the skull and left the dura intact.

**Primary hippocampal neuron culture and cell transfection**. The Sprague-Dawley rat pups (P0) was decapitated, and the hippocampus was dissected into the ice-cold dulbecco's modified eagle medium (DMEM) (catalog no. 11965-092, Gibco) containing 1% penicillin-streptomycin (P/S) (catalog no. 15140-122, Gibco). The hippocampus tissues were cut into small strips and digested with trypsin-EDTA (0.05%, catalog no. 25300-054, Gibco) for 20 min at 37 °C. The digestion reaction was terminated by DMEM supplemented with 10% heated-inactivated Fetal Bovine Serum (FBS) (catalog no. 10099-141, Gibco). Hippocampus tissues were triturated gently by the pipette (catalog no. 30-0138, BIO-LOGIX) for about 30 times and then stood for 2-3 min. After that, the collected neurons were plated onto poly-d-lysine (10 μg/mL, catalog no. A3890401, Gibco) - laminin (20 μg/mL, catalog no. 23017015, Invitrogen) coated coverslips, with the density adjusted to be $3 \times 10^5$.

We performed neurons transfections at DIV8 according to the manufacturer's instructions. In brief, we drop evenly 3 μl AAV2/9-hSyn-hChR2 (H134R) -EGFP-ER2-WPRE-pA with the titer of $1.34 \times 10^{13}$ into the neurons. At DIV14, we washed neurons with Modified Tyrode's Buffered Solution (catalog no. M301990, aladdin), and labelled the neurons with Cal-590 (catalog no. 20510, AAT Bioquest), to indicate calcium signal. In addition, pAA (DJD)-CaMKIIa-C1V1-TS-mCherry with the titer of $1 \times 10^{13}$ and pAAV(DJ)-hSyn-GCaMP6s with the titer of $1 \times 10^{13}$ were transfected into neurons, as the other paired opsin and calcium indicator.

**fCOAT focusing system design and integration of the calcium signal recording module**. The fCOAT system consists of three major modules (Supplementary Fig. 1, Supplementary Table 1): a modulation module, a feedback module, and a monitoring module. The modulation module tailors the incident wavefront to form a modulated pattern on the image plane. We used two combined continuous wave (CW) laser as sources, a 589-nm wavelength (Changchun New Industries Optoelectronics, MGL-FN-589-500mW) and a 488-nm wavelength (Coherent, Sapphire 488-300 CW CDRH). The two sources were both used for optogenetic stimulation and were modulated by an optical shutter (Daheng Optics, GCI-73 with GCI-7102M). Two 4-f systems (L1, L2 & L3, L4) and an iris (A1) were used to match the beam diameter of the CW laser to the effective area of the spatial light modulator (SLM; Meadowlark, A512-0532-P8), so that they can achieve optimum modulating efficiency. A half wave plate (P1) and a polarization beam splitter (PBS) were used to adjust the light intensity. The PBS and a polarizer (P2) were also used to select the modulated light and remove the unmodulated light. As for the other diffraction orders light, we used another iris (A2) to block them. A 4-f system (L5, L6) was used to conjugate the SLM plane to the rear pupil of the focusing objective (Obj.1; Olympus, PLN4X, N.A = 0.1).

The feedback module used another objective to collect the scattered light. To reach high collecting efficiency, we used high N.A objectives (Obj.2), either a ×20 (Nikon, CFI Plan Apochromat Lambda ×20, N.A = 0.75) or a ×40 (Nikon, CFI Plan Apochromat Lambda 40×C, N.A = 0.95) objective. We used a 550-nm shortpass dichroic mirror (DM2; Edmund, #69-215) to reflect 589 nm light, and an achromatic lens to image the scattered pattern to the multimode fiber (MMF) end face plane. The light collected by the MMF was directed to the photomultiplier tube (PMT; Hamamatsu, H7422P-40). The current signal output by the PMT was amplified by a preamplifier (Stanford Research Systems, SR570). We used a fast data acquisition card (DAQ; NI, USB-6366) to collect the trigger signal from the SLM and the corresponding amplified PMT signal.

The imaging module was used to obtain both the speckle, focus and the fluorescence images. Here we took 589 nm light for focusing and 488 nm light for imaging as an example. The imaging light was coupled to a single mode fiber (SMF). To guide the illumination light to the imaging system, we used a standard epi-fluorescence dichroic mirror (DM1; Semrock, FF506-Di03). The focusing light passed through it and reflected by DM2. Using another neutral density filter (OD4), the greatly attenuated light was imaged on the EMCCD camera (Andor, iXon 897 Ultra, 512 × 512 pixels, 16 μm pixel size). The emitted green fluorescence passed through both DM1 and DM2, and filtered by a bandpass filter (F1; 536/40) into the EMCCD camera.

The focusing light was switched using a flip mirror. The imaging light was switched to 560 nm using another CW laser (MPB Communication, 2RU-VFL-500-560-B1R) that was also coupled to the SMF. DM1 and DM2 were easily changed to suitable ones because we used magnetic lens holder. F1 can also be changed using a filter wheel.

**The working process of the fast multidither coherent optical adaptive technology (fCOAT)**. In our proposed method, the light field is segmented and modulated with different angular frequencies $\omega_i$, where $i$ represents different segments. The interference intensity signal can be expressed as below:

$$I(t) = \sum_{i=1}^{N}(C_i + 2A_iA_r cos(\omega_i t + \varphi_i)) \tag{1}$$

where $N$ represents the number of segments that are modulated at the same time, which is equal to half of the number of SLM segments in our method. $C_i = A_r^2 + A_i^2$ is a static intensity. The subscript $r$ represents the parameters of the static reference light. $\varphi_i$ represents the static phase induced by the scattering medium. We can take the Fourier transform of $I(t)$ to get phase curve as a function of modulation frequency $\omega$. Thus $\varphi_i$ corresponding to each $\omega_i$ can be obtained. By reverse loading the phase $\varphi_i$ to the SLM, the ith segment of the light field can achieve interference constructive, and the intensity of the target point can reach the maximum value. The other half of the SLM will go through the same process so that the whole light field can be compensated against the scattering (Supplementary Fig. 2a).

To demonstrate our parallel focusing procedure using binary modulation, we decided a standard to transform the continuous phase distribution to binary distribution first. We used 0 to replace the phase $0 \sim \pi/2$ and $3\pi/2 \sim 2\pi$, $\pi$ is used to replace $\pi/2 \sim 3\pi/2$ (Supplementary Fig. 2b). Because of using coherent technology, a reference light field was used to express the effect of the modulation phase. To create a stationary reference light field, the segments on the SLM (N × N) was separated into two halves. Each segment has an individual modulation frequency for the phase extraction after the Fourier transform of the collected time-intensity profiles. According to the Nyquist Sampling Law, the sampling frequency should be doubled the highest frequency $\omega_0$, which is $2\omega_0$. Half of the SLM segments ($N^2/2$) are modulated at first, then the other (Supplementary Fig. 2c), so the frequency resolution can be obtained as $2\omega_0/N^2$. Thus, to ensure sufficient frequency resolution, at least $N^2$ patterns should be added to the SLM during half modulation. A pattern sequence can be calculated by switching modulation phase between 0 and $\pi$ at different frequencies in these $N^2$ patterns. In order to improve the calculating accuracy, we set the lowest modulation frequency is half the highest frequency rather than 0 to reduce the influence of the few samples in the low frequency part. Therefore, the number of total loaded patterns is $4N^2$, $2N^2$ patterns each half. To further increase the modulation speed, we only used the segments that in the inscribed circle of the whole area of the SLM, which is also the area covered by the incident beam.

When the first half segments are modulated, the second half segments are set to 0 phase and act as the reference light field. After the Fourier transform changes the time-intensity profiles to phase-frequency profiles, the phases corresponding to different frequencies set above is found, and load them to the corresponding segments. When the first half is completed, the calculated results are loaded to the SLM as the reference light field when the second half is modulated. After loading half of the final phase, the signal intensity of the second half is improved significantly. The final correct wavefront is obtained by adding the results from the first and second half. All the phase calculated finally should also be binarized using the phase set proposed at the start of this part. The calculated phase pattern used in figures in the article are shown in Supplementary Fig. 7.

**Image analysis**. Images were processed using MATLAB or ImageJ. For focusing images and fluorescence images of neurons, the parts of interests are cropped out and tenfold bicubic interpolations were performed using "interp2" function in MATLAB. In addition, dark field background of the EMCCD was subtracted, the value of which is about 200, measured when the EMCCD was covered.

For speckle and focusing images, line profiles were extracted from selected interpolated images and then the PBR, which is defined as the ratio between the peak and the mean value of background of the signals, were calculated for wavefront shaping correction.

For calcium imaging data from neurons, circular regions of interest were placed over individual neurons to measure fluorescence using the "Plot Z-axis Profile" in ImageJ. The values of circular regions of blank areas near the target neurons were measured as background values and subtracted before calculating ΔF/F.

**Statistics and reproducibility**. For all the interval plots and box plots, the center lines are set to average values, the error bars and sample sizes are also described in figure captions. The individual data points used in the statistical analysis are provided in Source Data. Statistical significance was determined using GraphPad Prism 8.0.2 to perform a twotailed unpaired t-test. *P* values are provided in figure captions (Fig. 4 and Supplementary Fig. 4).

**Reporting summary**. Further information on research design is available in the Nature Portfolio Reporting Summary linked to this article.

## Data availability
The data supporting the findings of this study are available within the article, and its supplementary information and supplementary data files. Source data is located in Supplementary Data 1 and 2.

## Code availability
The MATLAB based wavefront compensation code are available in the Supplementary Software 1. The simulation code and other code supports the experiments of this study is also available in the public repository https://github.com/LiZhHan/fcoat with identifier https://doi.org/10.5281/ZENODO.7479131 (ref. [47]).

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

## Acknowledgements
This work was supported in part by National Natural Science Foundation of China (61735016), Natural Science Foundation of Zhejiang Province (LR22F050007), Key R&D Program of Zhejiang Province (2021C03001, 2020C03009), CAMS Innovation Fund for Medical Sciences (2019-I2M-5-057), China Postdoctoral Science Foundation

(2021M692816), Fundamental Research Funds for the Central Universities. The authors thank Lejia Hu for discussions on the implementation of the algorithm, and Jichao Du, Yue Zhu for the discussions on system setups. We also thank Lingxiao Gao, Xiao Xiao, Zizheng Wang and Sixia Zhong for the initial sample preparation, and Ao Deng for the help on the control program.

## Author contributions

K.S. and W.G. conceived the project. Z.L. led the project under the supervision of K.S. and W.G. Z.L. performed initial algorithmic simulations, designed and built the experimental setup, conducted the optical experiments. Z.L., X.D., and Y.X. conducted the biological experiments. X.D., Y.X., N.S. prepared the biological samples. Z.L. and Y.Z. analyzed the optical experimental data. Z.L., X.D., and R.L. analyzed the biological experimental data. X.L. and S.D. assisted with biological experiments and provided valuable guidance. All authors contributed to the discussions and preparation of manuscript.

## Competing interests

The authors declare no competing interests.
