## [Peer Review File · Communications Biology]

Reviewers' comments:

Reviewer #1 (Remarks to the Author):

In this paper, Li et al. report an adaptive wavefront correction method, fCOAT, to focus light in deep tissue. The method achieves adjustable focusing size up to the diffraction limit through 500-micron brain slices and overlapped mouse skulls. The authors also experimentally demonstrate focusing through live rat ears that lasts for more than 60 seconds when positioned away from blood vessels. Using the fCOAT techniques, the authors successfully achieve single-cellular optogenetic manipulation with improved stimulation efficiency.

Overall, I believe the authors present a compelling series of experiments with impressive results to show that the fCOAT system works great for optogenetics in deep tissues. This work should be of interest to the broad readership of Communications Biology. To improve the manuscript I suggest the following:

(1) According to Figure S2, the fCOAT system performs an FFT and then compresses the retrieved full phase to binary. What about the performance if this phase before binarization is used instead? The authors claim that the robustness is improved when using binary phase due to the higher tolerance of phase errors. Could the authors provide some evidence for this statement? For example, a simulation with random decorrelation with both the full and binary phases. Additional descriptions of Fig. S2 could also be helpful for readers to understand the workflow of the fCOAT system.

(2) Following the above question, can the authors comment on the quality degradation (5%? 10%? worse PBR) of the focusing spot using binary phase compared to full phase if there is no random scattering?

(3) The middle panel of the second row in Fig 4C. should not be labeled "baseline".

(4) There are numerous typos and grammar errors in the manuscript. For example:

The conclusion section (errors marked with [], correct ones in {}): "We report [a]{an} fCOAT focusing system that has super penetration depth and benefits for the non-invasive deep tissue optogenetics. The system demonstrates an ability to form a focus through highly scattering mouse brain tissue, mouse skulls, and dynamic live rat [ear]{ears}. The results [shows]{show} that the focus formed with a single fCOAT iteration [are] well-defined with high PBR, and [remain]{remains} valid ~60 s with live rat ear. Meanwhile, adjustable focus size is offered to match the requirements of coverage area. The [sytem]{system} achieves significant enhancements of stimulation responses for target neurons. The interference of nearby neurons is reduced due to the fCOAT system [suppresses]{suppressing} random scattering and [delivers]{delivering} power more efficiently to the focus. The advantages of adjustable focus, high speed and robustness, high penetration depth make the fCOAT system [beome]{become} a promising tool for noninvasive deep tissue optogenetics manipulation. It also paves the way for precise stimulations at unprecedented depth that is [benefit]{beneficial} to in vivo neural studies involving highly turbid samples including, but not limited to, neuroscience and brain science."

Title of Fig. 4 "The manipulation of fCOAT system on neural activities" should probably be "the manipulation of neural activities using the fCOAT system".

Proofreading and fixing these errors could improve the readability of the manuscript.

Reviewer #2 (Remarks to the Author):

Non invasive approaches are advantages over invasive methods as they eliminate the need for surgery for insertion of device. However, such tether systems are subject to motion artifacts, which significantly degrade signal integrity. This is a major weakness and diminished author's

enthusiasm. Moreover, conventional fiber approaches have been well established and I don't see any benefit that the proposed approach could offer. My suggestion is authors should characterize motion artifact effects systematically.

Reviewer #3 (Remarks to the Author):

This paper introduces a fast wavefront shaping technique with few iterations, aimed at overcoming dynamic scattering. The technique is then applied to optogenetic stimulation and enhanced excitation for stronger fluorescent signals. The technique is technically novel, and the biological application is of general interest in the field of biophotonics, wavefront shaping and neuroscience. The work, however, needs further clarification and explanation before it can be considered for acceptance of publication.

Major comments:

1. A major concern of the work is the lack of clarity in the wavefront shaping algorithm.

1.1 In the abstract, it is claimed that only one iteration of wavefront shaping was performed. Despite the fast wavefront shaping, it is not clear that it can overcome dynamic scattering based on the data. During the dynamic process, I would expect that the algorithm will continuously compensate for drift in focus so that a focus can be maintained for a longer period of time. Rather than just one-time compensation, continuous photo-stimulation in a longer time window is more helpful, desirable, and interesting for broader biological applications.

1.2 The algorithm appears to only be capable of generating arbitrary SINGLE foci in the field of view. The majority of the data demonstrate that they tend to focus on the center of the field of view. Would it be possible to target focusing on a certain spot or multiple spots? How, if so?

1.3 Since only binary modulation is required for the algorithm to work, can the SLM be replaced by a DMD?

1.4 In addition to the mathematical description of the algorithms, the authors should include a brief explanation of why the algorithm can converge to a spot. The most challenging aspect of reproducing the work will be the lack of clarity in this section. The algorithm code should be publicly accessible for public dissection in an open-source repository.

1.5 A detailed data processing procedure needs to be added to Fig S2. Is there a background removal or denoising step being added in any of the procedures?

2. How much laser power is applied directly to the samples under the objectives? Only the direct output power of the laser is mentioned in the paper. It seems that the power level is too low to activate optogenetic systems if it's only a few hundred micro-W right after the laser. Can the authors clarify this?

3. In all focusing figures (fig 1b, 2b, 2g, fig 3b 3f, fig 3c, 3f, fig 5), what phase compensation patterns are used? To visually determine the level of scattering/aberration being corrected, they need to be added.

4. In line 198, can the authors explain why a 20 μm beam is used for optogenetics? With neuron sizes typically around 10 μm , would 20 μm be too large? Furthermore, despite the authors' claim that ref 31 has a larger focus, this size is already close to the spot size in ref 31. The authors need to comment on it.

5. How does the author explain fig 4j and 5b, where there is no obvious signal increase in traces after focusing? Does this mean the dynamic compensation failed?

6. I would suggest tuning down the claim "non-invasive" in the title since the experiments so far are not fully non-invasive.

Minor comments:

7, Several typos here and there that need to be corrected.

7,1 For example, in line 43, "ten times" should be "tens of".

7,2 In line 60, "argrithims" should be "algorithm"

7.3 In lines 131, 143, and 151, elsewhere, "serious scattering" to "severe scattering".

7,4 Fig. 3(a) "used mouse" needs to be replaced with a more proper word. "(top) and (bottom)" should be "(left) and (right)".

8, In line 130, what is the effective NA being used? Here is how the 3.6 μm is determined as the diffraction-limited spot.

Dear reviewers,

We would like to express our sincere gratefulness to the reviewers for their insightful comments and suggestions on our manuscript. We have revised the manuscript according to reviewers' comments as follows:

Reviewer #1

In this paper, Li et al. report an adaptive wavefront correction method, fCOAT, to focus light in deep tissue. The method achieves adjustable focusing size up to the diffraction limit through 500-micron brain slices and overlapped mouse skulls. The authors also experimentally demonstrate focusing through live rat ears that lasts for more than 60 seconds when positioned away from blood vessels. Using the fCOAT techniques, the authors successfully achieve single-cellular optogenetic manipulation with improved stimulation efficiency.

Overall, I believe the authors present a compelling series of experiments with impressive results to show that the fCOAT system works great for optogenetics in deep tissues. This work should be of interest to the broad readership of Communications Biology. To improve the manuscript I suggest the following:

Reply:

Thanks for your comments.

We are very appreciative of your agreement on the significance of our work.

(1) According to Figure S2, the fCOAT system performs an FFT and then compresses the retrieved full phase to binary. What about the performance if this phase before binarization is used instead? The authors claim that the robustness is improved when using binary phase due to the higher tolerance of phase errors. Could the authors provide some evidence for this statement? For example, a simulation with random decorrelation with both the full and binary phases. Additional descriptions of Fig. S2 could also be helpful for readers to understand the workflow of the fCOAT system.

Reply:

Thanks for your comments.

a. The performance of full phase and binary phase modulation.

According to the article "Focusing light inside dynamic scattering media with millisecond digital optical phase conjugation" published by Liu. Y et al. in 2017 (ref. 13), the peak intensity achieved by binary phase modulation is about 40% of that achieved using full phase modulation. We simulated the scattering compensation comparing full and binary phase modulation, which obtained the same results from ref. 13, as shown in **Reply Fig. 1**.

Reply Fig. 1. Comparison of full and binary phase modulation. (a) Comparison of the normalized intensity of the formed focus in 25 random scattering simulations between full phase modulation and binary phase modulation. (b) Statistical results of the normalized intensity in (a). The error bars represent SE of the 25 measured results.

b. Improved robustness.

The robustness using binary phase modulation is reflected in two aspects, the ability to form a well-defined focus and the tolerance of phase errors after the focus is formed. Due to only one-bit data required for binary phase modulation, while the full phase modulation needs eight-bit data for each SLM pixel, and the high refresh rate of FLC-SLM is about eight times of ordinary SLMs. This means each phase loading of binary phase modulation only experiences one-eighth the number of times decorrelation of full phase modulation, which ensures the method can form a focus faster and more stable while full phase modulation cannot even form a tight focus (**Reply Fig. 2**). The intensities are normalized based on the focus intensity of binary phase modulation.

Reply Fig. 2. Results of focusing when facing dynamic scattering using full and binary phase modulation.

As for the tolerance of phase errors induced by decorrelation of scattering phase pattern, we ran a simulation to confirm this idea. Our proposed method calculates the

phase of different SLM segments according to the scattering pattern, the decorrelation will cause the change of calculated compensation phase (**Reply Fig. 3a**). Here we perform the decorrelation of compensation phase, which is corresponding to the phase results calculated for different scattering pattern. Different from full phase modulation, due to the phase compression of binary phase modulation, not all decorrelation phases will cause the change of 0 or π phase (**Reply Fig. 3b**). The figure below shows the decorrelation induced intensity decreasing of the focus.

Reply Fig. 3. Comparison of intensity decreasing when modulation phase changes. (a) The results between full phase modulation and binary phase modulation when the modulation patterns are decorrelated. The top row shows the initial phase pattern and corresponding focus image. The bottom row shows the final phase pattern after 50 times decorrelation and corresponding focus image. (b) Intensity decreasing profiles when the calculated phase changes due to the scattering decorrelation. The intensity values are normalized to the initial focus intensity values.

The modulation patterns are gradually decorrelated, which is the simulation of the calculated phase change due to different scattering patterns. Each time decorrelation causes the calculated full phase modulation patterns make differences, while the binary phase modulation patterns are not necessarily affected, which means a binary phase pattern corresponds to a variety of full phase patterns. As shown in the line graph on the right, although the overall intensity attenuation times are basically the same, low degree of scattering decorrelation does not affect the final calculated phase, and corresponding focus intensity decreases slowly, which gives binary phase modulation more possibilities to maintain focus. Thus, we claim that the tolerance of phase errors of binary phase modulation is higher and the system robustness is improved compared to full phase modulation.

When decorrelating the scattering phase pattern, the binary phase modulation also maintains the focus better than full phase modulation. As shown in **Reply Fig. 4**, when the two modulation modes experience same times of decorrelation, binary phase mode has more possibilities to maintain the focus intensity.

Reply Fig. 4. Results of focus intensity decreasing when the scattering patterns are decorrelated. The intensity values are normalized to the initial focus intensity values.

The intensities recorded are normalized based on the respective intensities of focus formed by the two modes themselves. We have also revised some inaccurate statements in the article.

In line 191 and 192, we revised the original text to “the robustness of the fCOAT system is higher than the full phase modulation and the focus have the ability to maintain longer during the decorrelation of the living tissue”.

In line 206, we revised the original text to “The robustness of fCOAT system mainly arises from the high modulation speed and the higher tolerance of phase errors”, in line 209, the text is revised to “The ability of the formed focus through live rat ear is a typical example of these two features”.

In line 211-215, we changed the text to “the intersection of the phase pattern correspond to each focus is the majority of the final phase pattern to generate a large focus. The binary phase modulation has lower PBR than full phase modulation, but the binary pattern leads to smaller conflicts and the gap with PBR of full phase modulation will not be further widened. Moreover, the signal enhancement maintains at higher than an order of magnitude although lower than small focus”.

We added extra text to the legend of Fig. S2, as follows:

“The phase patterns are quickly loaded onto the SLM in turn, resulting in a different modulation frequency for each SLM segments. The time-domain signal collected from the target position is transformed to frequency-domain. The phase of each frequency is loaded to the corresponding SLM segments. After the first loop, half of the total SLM segments are loaded corresponding phase, and then the second loop to determine the phase of the rest segments.”

(2) Following the above question, can the authors comment on the quality degradation (5%? 10%? worse PBR) of the focusing spot using binary phase compared to full phase if there is no random scattering?

Reply:

Thanks for your comments.

The FLC-SLM only works as an ordinary mirror when there is no random scattering, and is independent of full phase or binary phase.

(3) The middle panel of the second row in Fig 4C. should not be labeled "baseline".

Reply:

Thanks for your comments.

We changed the word "baseline" to "base"¹.

(4) There are numerous typos and grammar errors in the manuscript. For example:

The conclusion section (errors marked with [], correct ones in {}): "We report [a]{an} fCOAT focusing system that has super penetration depth and benefits for the non-invasive deep tissue optogenetics. The system demonstrates an ability to form a focus through highly scattering mouse brain tissue, mouse skulls, and dynamic live rat [ear]{ears}. The results [shows]{show} that the focus formed with a single fCOAT iteration [are] well-defined with high PBR, and [remain]{remains} valid ~60 s with live rat ear. Meanwhile, adjustable focus size is offered to match the requirements of coverage area. The [sytem]{system} achieves significant enhancements of stimulation responses for target neurons. The interference of nearby neurons is reduced due to the fCOAT system [suppresses]{suppressing} random scattering and [delivers]{delivering} power more efficiently to the focus. The advantages of adjustable focus, high speed and robustness, high penetration depth make the fCOAT system [beome]{become} a promising tool for noninvasive deep tissue optogenetics manipulation. It also paves the way for precise stimulations at unprecedented depth that is [benefit]{beneficial} to in vivo neural studies involving highly turbid samples including, but not limited to, neuroscience and brain science."

Title of Fig. 4 "The manipulation of fCOAT system on neural activities" should probably be "the manipulation of neural activities using the fCOAT system".

Proofreading and fixing these errors could improve the readability of the manuscript.

Reply:

Thanks for your comments.

We carefully checked and corrected the text for grammar and spelling errors, thanks for your carefully review.

Reviewer #2

Non invasive approaches are advantages over invasive methods as they eliminate the need for surgery for insertion of device. However, such tether systems are subject to motion artifacts, which significantly degrade signal integrity. This is a major weakness and diminished author's enthusiasm. Moreover, conventional fiber approaches have been well established and I don't see any benefit that the proposed approach could offer. My suggestion is authors should characterize motion artifact effects systematically.

Reply:

Thanks for your comments.

Motion artifacts is more common in imaging systems, mainly due to the animal's heartbeat or heart pulsation and result in a loss of subcellular details, image position offset and decrease of the signal-to-noise ratio (SNR). To overcome this issue in multiphoton imaging systems, electrocardiogram (ECG)-triggered scanning strategy has been used^{2,3}. By synchronizing imaging scans to the cardiac cycle in ECG triggered strategy, the resolution of cellular structures can be significantly improved. However, considering the optical memory effect^{4,5}, the image intensity can be described as the convolution of the excitation focus PSF and the object, and the PSF of the focus is approximately static within the memory effect range, which is also the fields of view of the obtained image. The focus should be formed using wavefront shaping or adaptive optics strategies first, and image refactoring, which will be influenced by motion artifacts, is performed after collecting image intensity. With or without ECG triggered strategy to reduce motion artifacts, the collected image intensity at each point in the image is basically the same, but the results after refactoring will be different due to motion artifacts, which means that the signal collection at each point and the ECG triggered strategy are two separated processes.

In imaging applications, wavefront shaping, or adaptive optics techniques, is combined with signal collection part to compensate for low-order aberrations induced by biological tissues that reduce the focus quality. Similarly, our proposed wavefront shaping method is designed to quickly compensate for both low- and high-order aberrations and scattering induced by biological tissues to form a tight focus and is also not influenced by motion artifacts. The problem that is easily confused with motion artifacts encountered during wavefront shaping processes is dynamic scattering. Different from motion artifacts that caused by the intermittent and violent shake or move of the objects, dynamic scattering is mainly induced by the fluctuations caused by blood flow that keep appearing. We have demonstrated the ability of our method to overcome this problem in the article, as shown in Fig. 3.

Moreover, although conventional fiber approaches have been widely used for optical stimulation^{6,7}, the spatial resolution is greatly restricted to larger than hundreds of microns because light emerging from an optical fiber, whether single-mode or multi-mode fiber, is

diverging. Our proposed method increases this resolution to few microns - a cellular level of resolution that is difficult to achieve with conventional fiber approaches and benefits a lot for high-precision optogenetics.

Reviewer #3

This paper introduces a fast wavefront shaping technique with few iterations, aimed at overcoming dynamic scattering. The technique is then applied to optogenetic stimulation and enhanced excitation for stronger fluorescent signals. The technique is technically novel, and the biological application is of general interest in the field of biophotonics, wavefront shaping and neuroscience. The work, however, needs further clarification and explanation before it can be considered for acceptance of publication.

Major comments:

1. A major concern of the work is the lack of clarity in the wavefront shaping algorithm.

1.1 In the abstract, it is claimed that only one iteration of wavefront shaping was performed. Despite the fast wavefront shaping, it is not clear that it can overcome dynamic scattering based on the data. During the dynamic process, I would expect that the algorithm will continuously compensate for drift in focus so that a focus can be maintained for a longer period of time. Rather than just one-time compensation, continuous photo-stimulation in a longer time window is more helpful, desirable, and interesting for broader biological applications.

Reply:

Thanks for your comments.

Almost all methods can be modified to continuous compensation mode by adjusting some parameters, like the methods reported in ref. 15 & 28. The core of the modification is to choose some of the whole segments, recalculate the phase of these segments and load them on the SLM again. Comparing to totally recalculate the whole phase pattern, the continuous working mode maintains the focus because only a few segments are

Reply Fig. 5. The simulation of iteration mode of fCOAT system. (a) The intensities of target points during continuous working mode. Each calculation determines only one-eighth of the whole SLM. (b) Modulation frequency distribution of each SLM segment in one-time compensation mode (two parts) and continuous compensation mode (eight parts) used to obtain results in (a).

recalculated, which can be regarded as phase perturbation. The fraction of segments selected for recalculation depends on the decorrelation speed of the dynamic medium. Here we give a simulation of the continuous compensation mode of the proposed method to demonstrate the capability of the system (**Reply Fig. 5a**), where one-eighth part of the whole SLM segments are selected to recalculate to compensate for the drift in focus (**Reply Fig. 5b**).

After each part of the SLM iteration is completed, we make the scattering phase decorrelate slowly to see the formed focus intensity better. During the phase calculation part, we let the scattering phase decorrelate faster to quickly reduce the focus intensity to show the ability of the algorithm for continuously compensation for drift in focus. We select data collected in three simulations to show the performance.

We put this part in Supplementary Fig. S7 as extra features.

1.2 The algorithm appears to only be capable of generating arbitrary SINGLE foci in the field of view. The majority of the data demonstrate that they tend to focus on the center of the field of view. Would it be possible to target focusing on a certain spot or multiple spots? How, if so?

Reply:

Reply Fig. 6. Experimental results of focus on different positions. (a) Focusing at different locations by moving the collection point. The insets show the speckle and focus pattern before and after wavefront compensation, the intensities of speckle patterns are enhanced five times. Scale bars are shown in the figures. (b) The statistical data of PBR of focus formed at different locations. The results are normalized using the average value of the PBR of focus of all different locations. (c) The statistical data of the distance between focus position and the expected position of different focus locations. Error bars in (b) and (c) represent the SE of ten measurements taken at random different locations on brain slices.

Thanks for your comments.

The focusing results depend on the position of the acquisition point of the feedback signals in our proposed method. We add experiments to target focusing on certain spots here.

The acquisition points are set to nine different points of 2 mm spacing. Due to the acquisition plane is conjugated to the camera and the effect of the 20× objective, the actual focus displacement each time should be 100 μm (**Reply Fig. 6a**). The statistical data of focusing PBR and focus drifts of ten experiments are shown together (**Reply Fig. 6b, c**). We can find that the no significant changes of focusing performance are observed.

We also put this part in Supplementary Fig. S7 as extra features.

As for focusing on multiple spots, signals from different spots need to be collected at the same time, which is more suitable for camera collection and greatly reduces the modulation speed due to the relatively low frame rate of common cameras. Since our system is set for fast focusing that tends to use faster collection devices, we run a simulation to verify that focusing on multiple spots can be achieved (**Reply Fig. 7**). From the results below, we can see that when focusing on multiple spots, the positions of formed focus still locate at our target position and only the focus intensity is reduced because it needs to be divided into several spots.

Reply Fig. 7. Simulation results of focusing on multiple spots. The intensity values are normalized to the mean intensity values of single focus (labeled as Left and Right).

1.3 Since only binary modulation is required for the algorithm to work, can the SLM be replaced by a DMD?

Reply:

Thanks for your comments.

We think that DMD is not suitable to replace the FLC-SLM here mainly because our

proposed method is based on phase modulation rather than intensity modulation.

Our proposed method is based on the interference of the incident coherent light. Half of the SLM segments illuminated by the incident light are kept static when the other half segments are calculating the half of the phase pattern. So that the static half of the incident light can perform as the reference light to obtain the intensity changes during the phase calculation. The phase-only modulation induced by the FLC-SLM is rather suitable for this situation. DMD is commonly known as an intensity modulation device, converting it to phase modulation device is much more complicated. Using intensity modulation, the segments that are not constructive for the PBR of the target point are removed. This feature not only reduces modulation efficiency but is also inconsistent with the principle of the proposed method.

1.4 In addition to the mathematical description of the algorithms, the authors should include a brief explanation of why the algorithm can converge to a spot. The most challenging aspect of reproducing the work will be the lack of clarity in this section. The algorithm code should be publicly accessible for public dissection in an open-source repository.

Reply:

Thanks for your comments.

In our proposed method, the light field is segmented and modulated with different angular frequencies ω_i , where i represents different segments. The interference intensity signal can be expressed as below:

$$I(t) = \sum_{i=1}^N (C_i + 2A_i A_r \cos(\omega_i t + \varphi_i))$$

Where N represents the number of segments that are modulated at the same time, which is equal to half of the number of SLM segments in our method. $C_i = A_r^2 + A_i^2$ is a static intensity. The subscript r represents the parameters of the static reference light. φ_i represents the static phase induced by the scattering medium. We can take the Fourier transform of $I(t)$ to get phase curve as a function of modulation frequency ω . Thus φ_i corresponding to each ω_i can be obtained. By reverse loading the phase φ_i to the SLM, the i th segment of the light field can achieve interference constructive, and the intensity of the target point can reach the maximum value. The other half of the SLM will go through the same process so that the whole light field can be compensated against the scattering.

We also added the above part to the **Methods** part in the main text.

We have provided a full code availability statement in the manuscript. The MATLAB based wavefront compensation code are available in the Supplementary Software. The LabVIEW based hardware control codes and MATLAB based analysis and simulation codes are available from the corresponding authors upon reasonable request.

1.5 A detailed data processing procedure needs to be added to Fig S2. Is there a background removal or denoising step being added in any of the procedures?

Reply:

Thanks for your comments.

In the data acquisition procedure, the sample rate of the PMT was set to 2MHz, and it took 400 samples per phase pattern, as described in the legend to Supplementary Fig. S3. This way of sampling such multiple times is an average way to denoise. The signal amplifying was operated by SR570 amplifier. Then the Fourier transform and phase calculation. No other data processing procedure like background removal or denoising step needs to be added.

2. How much laser power is applied directly to the samples under the objectives? Only the direct output power of the laser is mentioned in the paper. It seems that the power level is too low to activate optogenetic systems if it's only a few hundred micro-W right after the laser. Can the authors clarify this?

Reply:

Thanks for your comments.

Sorry for the lack of clarity of this point you mentioned here. The laser power used for optogenetic mentioned in the article is all measured after the objective, as mentioned in line 274, "before scattered by brain slices". With few hundred micro-W after the objective, the power density at the target point reaches more than few hundred watts per square centimeter, and more than tens of watts per square centimeter even after the scattering, which is enough for the optogenetic activation systems.

We have corrected this issue in the article, in line 120, 176, 225 and 255, we added "after the focusing objective", thanks for your reminding.

3. In all focusing figures (fig 1b, 2b, 2g, fig 3b 3f, fig 3c, 3f, fig 5), what phase compensation patterns are used? To visually determine the level of scattering/aberration being corrected, they need to be added.

Reply:

Thanks for your comments.

We add the used compensation patterns in focusing figures in Supplementary Fig. S6 (Reply Fig. 8), as shown below.

Reply Fig. 8. Phase compensation patterns used in focusing figures in the article. The labels are the same as in the article.

4. In line 198, can the authors explain why a 20 μm beam is used for optogenetics? With neuron sizes typically around 10 μm , would 20 μm be too large? Furthermore, despite the authors' claim that ref. 31 has a larger focus, this size is already close to the spot size in ref. 31. The authors need to comment on it.

Reply:

Thanks for your comments.

As shown in Fig. 4c in our article, a focus with FWHM spot size of $\sim 3.6 \mu\text{m}$ can only cover part of the neuron, while in Fig. 4i (with scale bar) that a focus with FWHM spot size of $\sim 10 \mu\text{m}$ can cover the whole neuron better and this size was not too large for neurons.

In ref. 31, it mentioned that “achieve a focus with an average FWHM spot size of 27.4

μm ". Here we chose to expand the focus to $\sim 20 \mu\text{m}$ FWHM mainly to demonstrate the ability of the system. We experimentally demonstrate that our system can adjust the focus FWHM spot size from $\sim 3.6 \mu\text{m}$ to $\sim 20 \mu\text{m}$, and the experiments used the size of $\sim 3.6 \mu\text{m}$ and $\sim 10 \mu\text{m}$ FWHM, while the technique mentioned in ref. 31 can only obtain larger focus (FWHM larger than $\sim 20 \mu\text{m}$) and have less precision when facing smaller neurons.

5. How does the author explain fig 4j and 5b, where there is no obvious signal increase in traces after focusing? Does this mean the dynamic compensation failed?

Reply:

Thanks for your comments.

In Fig. 4j and 5b, the red lines represent the activation results using formed focus and the focus was located only at the position of the neuron marked as #1. It is obvious that after being stimulated by the formed focus, the $\Delta F/F$ of #1 neuron was significantly different from the other three neurons, which is because the focus was precise enough to select a certain neuron. Only the target neuron was stimulated with focus and produced a calcium signal change that closely followed the stimulation light, the other neurons were still stimulated by speckle light field and would not be directly and regularly influenced by the stimulation light, resulting in no obvious changes in calcium signal.

6. I would suggest tuning down the claim "non-invasive" in the title since the experiments so far are not fully non-invasive.

Reply:

Thanks for your comments.

Conventional optogenetic manipulations currently require the surgical implantation of invasive optical fibers for light delivery below the most superficial brain regions⁸, while our proposed method eliminates the need for surgery for insertion of devices to directly contact with the targets. Moreover, our focusing method does not require any artificial probe as invasive point source, and this kind of strategies are commonly referred as noninvasive strategies^{9,10}. Therefore, we consider our system is noninvasive.

Our method allows for an optical focus to be formed noninvasively with the ability to freely move the adjustable focus within the target plane to target different regions of interest, we believe that our method will contribute to the development of noninvasive optogenetic stimulation.

In conclusion, we think that the current title can express our topic better comparing to tuning down the "non-invasive".

Minor comments:

7, Several typos here and there that need to be corrected.

7,1 For example, in line 43, “ten times” should be “tens of”.

7,2 In line 60, “argrithims” should be “algorithm”

7.3 In lines 131, 143, and 151, elsewhere, “serious scattering” to “severe scattering”.

7,4 Fig. 3(a) “used mouse” needs to be replaced with a more proper word. “(top) and (bottom)” should be “(left) and (right)”.

Reply:

Thanks for your comments.

We carefully checked and corrected the text for grammar and spelling errors, thanks for your carefully review.

8, In line 130, what is the effective NA being used? Here is how the 3.6 um is determined as the diffraction-limited spot.

Reply:

Thanks for your comments.

Here we used an objective with low NA (Olympus, PLN4X, NA=0.1), in line 130-131, we let the light fulfill the rear pupil of the objective and used the whole NA. According to the diffraction limit formula:

$$FWHM = 0.61 \frac{\lambda}{NA}$$

Using 589 nm laser, and 0.1 NA, the FWHM was estimated to be 3.5929 μm.

References

1. Li Q, DeBeaubien NA, Sokabe T, Montell C. Temperature and Sweet Taste Integration in *Drosophila*. *Curr Biol* **30**, 2051-2067 e2055 (2020).
2. Streich L, *et al.* High-resolution structural and functional deep brain imaging using adaptive optics three-photon microscopy. *Nat Methods* **18**, 1253-1258 (2021).
3. Paukert M, Bergles DE. Reduction of motion artifacts during in vivo two-photon imaging of brain through heartbeat triggered scanning. *J Physiol* **590**, 2955-2963 (2012).
4. Schott S, Bertolotti J, Leger JF, Bourdieu L, Gigan S. Characterization of the angular memory effect of scattered light in biological tissues. *Opt Express* **23**, 13505-13516 (2015).
5. Papadopoulos IN, Jouhannau J-S, Poulet JFA, Judkewitz B. Scattering compensation by focus scanning holographic aberration probing (F-SHARP). *Nature Photonics* **11**, 116-123 (2016).
6. Sych Y, Fomins A, Novelli L, Helmchen F. Dynamic reorganization of the cortico-basal ganglia-thalamo-cortical network during task learning. *Cell Rep* **40**, 111394 (2022).
7. Sych Y, Chernysheva M, Sumanovski LT, Helmchen F. High-density multi-fiber photometry for

- studying large-scale brain circuit dynamics. *Nat Methods* **16**, 553-560 (2019).
8. Zhang F, *et al.* Optogenetic interrogation of neural circuits: technology for probing mammalian brain structures. *Nat Protoc* **5**, 439-456 (2010).
 9. Wu T, Dong J, Gigan S. Non-invasive single-shot recovery of a point-spread function of a memory effect based scattering imaging system. *Opt Lett* **45**, 5397-5400 (2020).
 10. Wang D, Sahoo SK, Zhu X, Adamo G, Dang C. Non-invasive super-resolution imaging through dynamic scattering media. *Nat Commun* **12**, 3150 (2021).

Reviewers' comments:

Reviewer #1 (Remarks to the Author):

The authors addressed all my concerns and the manuscript is much improved. I think the article in its current version is acceptable for publication.

Reviewer #2 (Remarks to the Author):

The rationale for achieving a spatial resolution of below 10um is limited. First of all, most optogenetic experiments do not require a few um level of spatial resolution unless you aim at a single individual neuron (this experiment does not provide any scientific significance). Moreover, expression of light sensitive opsins to the targeted region does not require such a high level of spatial resolution.

In non-vasive approaches, securing a light source to the interface is critical as deviation of 'light delivery to the targeted region' would be significant, although the deviation at the region of light source is negligible. Even if a fiber optic cable is secured to the skull, such a light source would be displaced(or migrated) during animal motion and the level of displacement would be significant.

Reviewer #3 (Remarks to the Author):

My previous comments except # 6 are all satisfactorily addressed.

My disagreement still resides with the authors' claim of "non-invasive focusing". The demonstrated work is both biological and optically invasive.

To explain this further, the authors demonstrate the focusing THROUGH extracted ex vivo brain slice and excited cultured neurons that are not embedded in but outside of the brain slice. Such a design is not yet non-invasive due to the fact that the camera can already directly image neurons from the other side.

In the article, the authors referred to fiber photometry, where insertion into tissue is required for optogenetics. This comparison is not completely fair. The reason for this is that fiber photometry is typically used in practical settings where the brain is intact and in vivo. In this study, the authors were unable to demonstrate photostimulation on in intact brains, still requiring cutting of the brain and separately placing labeled and cultured neurons outside/behind the brain slices (unlabeled), and were unable to stimulate neurons INSIDE the brain or thick brain slices. In this aspect, this work itself is biologically invasive.

Furthermore, I would like to discuss the optical design aspect of the work in order to explain why the experimental design is also invasive. This work has an experimental design typically found in imaging through scattering medium experiments, where the camera has direct access to the real target in a transmission setup (camera and photostimulation laser/SLM are on the two different sides of the sample). It is well known in the field that such a design does not constitute a true non-invasive method [1], so the most representative non-invasive scattering experiments design their experimental setup in a reflection mode [1,2], which means that the fluorescence camera, SLM, and laser are all on the same side of the scattering sample, making it applicable for non-invasive focus. Furthermore, even with current transmission designs, where cultured neurons are already exposed outside thick tissue, fiber photometry can also excite neurons in a "non-invasive" manner according to the authors' argument, which is not fair and not convincing.

I hope the authors understand that the real purpose of non-invasive optogenetics of deep tissue is to stimulate neurons INSIDE thick tissues. Due to this motivation, the authors can improve their future experimental design in order to have a true non-invasive sample where the neurons are within the thick tissue and the camera cannot directly image them [3], or a reflection setup that is closer to the actual in vivo imaging scenario in which all optical elements can only be placed on

one side of the brain.

I strongly suggest that the authors remove the non-invasive claim from the title and abstract of the paper. A discussion of the potential strategies toward a non-invasive approach would add a significant amount of value to this work, I believe. If all of the above concerns can be addressed accordingly, I would suggest accepting the work without further concern.

[1] Bertolotti, Jacopo, et al. "Non-invasive imaging through opaque scattering layers." *Nature* 491.7423 (2012): 232-234.

[2] Boniface, Antoine, Jonathan Dong, and Sylvain Gigan. "Non-invasive focusing and imaging in scattering media with a fluorescence-based transmission matrix." *Nature communications* 11.1 (2020): 1-7.

[3] Ruan, Haowen, et al. "Deep tissue optical focusing and optogenetic modulation with time-reversed ultrasonically encoded light." *Science advances* 3.12 (2017): eaao5520.

Dear reviewers,

We would like to express our sincere gratefulness to the reviewers for their insightful comments and suggestions on our manuscript. We have revised the manuscript according to reviewers' comments as follows.

Reviewer #1:

The authors addressed all my concerns and the manuscript is much improved. I think the article in its current version is acceptable for publication.

Reply:

Thank you for your comments.

Reviewer #2:

The rationale for achieving a spatial resolution of below 10um is limited. First of all, most optogenetic experiments do not require a few um level of spatial resolution unless you aim at a single individual neuron (this experiment does not provide any scientific significance). Moreover, expression of light sensitive opsins to the targeted region does not require such a high level of spatial resolution.

Reply:

Thanks for your thoughtful comments.

The size of the focus formed by our strategy can be adjusted (**Fig. 2b** and **Fig. 3f**). The spatial resolution of below 10 μm is the highest resolution while maintaining deep penetration depth and this demonstrate the ability of the system to perform high precision optogenetic that aims at individual neurons, which is more common in multiphoton mode^{1, 2}. Moreover, although expression of light sensitive opsins does not require such high level of spatial resolution, this level of resolution is required in precise light stimulation.

In non-vasive approaches, securing a light source to the interface is critical as deviation of 'light delivery to the targeted region' would be significant, although the deviation at the region of light source is negligible. Even if a fiber optic cable is secured to the skull, such a light source would be displaced(or migrated) during animal motion and the level of displacement would be significant.

Reply:

Thanks for your thoughtful comments.

This light source displacement caused by animal motion was considered as system

robustness and discussed in the article at several parts. First, in **Fig. 2** and **Fig. 3**, we have shown the stable focusing results through different scattering medium. The statistical data is obtained from focus at different positions of the scattering medium. As shown in the figures, the focus formed at different positions are similar, which means the focus results after animal motions are basically the same. Especially, the focus displacements data is shown in **Fig. 2e**, the level of displacements are mostly less than 1 μm , showing that the position of the formed focus (referred as the light source on the interface) is very stable.

Moreover, in **Supplementary Fig. S7a**, we show the results of the continuous working mode of the system. When facing animal motion that can lead to significant displacement of the focus, the system can compensate for the displacement fast.

These two features are critical in non-invasive approaches and make sure the animal motion can be overcome.

Reviewer #3:

My previous comments except # 6 are all satisfactorily addressed.

My disagreement still resides with the authors' claim of "non-invasive focusing". The demonstrated work is both biological and optically invasive.

To explain this further, the authors demonstrate the focusing THROUGH extracted ex vivo brain slice and excited cultured neurons that are not embedded in but outside of the brain slice. Such a design is not yet non-invasive due to the fact that the camera can already directly image neurons from the other side.

In the article, the authors referred to fiber photometry, where insertion into tissue is required for optogenetics. This comparison is not completely fair. The reason for this is that fiber photometry is typically used in practical settings where the brain is intact and in vivo. In this study, the authors were unable to demonstrate photostimulation on intact brains, still requiring cutting of the brain and separately placing labeled and cultured neurons outside/behind the brain slices (unlabeled), and were unable to stimulate neurons INSIDE the brain or thick brain slices. In this aspect, this work itself is biologically invasive.

Furthermore, I would like to discuss the optical design aspect of the work in order to explain why the experimental design is also invasive. This work has an experimental design typically found in imaging through scattering medium experiments, where the camera has direct access to the real target in a transmission setup (camera and photostimulation laser/SLM are on the two different sides of the sample). It is well known in the field that such a design does not constitute a true non-invasive method [1], so the most representative non-invasive scattering experiments design their experimental setup in a reflection mode [1,2], which means that the fluorescence camera, SLM, and laser are all on the same side of the scattering sample, making it applicable for non-invasive focus.

Furthermore, even with current transmission designs, where cultured neurons are already exposed outside thick tissue, fiber photometry can also excite neurons in a "non-invasive" manner according to the authors' argument, which is not fair and not convincing.

I hope the authors understand that the real purpose of non-invasive optogenetics of deep tissue is to stimulate neurons INSIDE thick tissues. Due to this motivation, the authors can improve their future experimental design in order to have a true non-invasive sample where the neurons are within the thick tissue and the camera cannot directly image them [3], or a reflection setup that is closer to the actual in vivo imaging scenario in which all optical elements can only be placed on one side of the brain.

I strongly suggest that the authors remove the non-invasive claim from the title and abstract of the paper. A discussion of the potential strategies toward a non-invasive approach would add a significant amount of value to this work, I believe. If all of the above concerns can be addressed accordingly, I would suggest accepting the work without further concern.

[1] Bertolotti, Jacopo, et al. "Non-invasive imaging through opaque scattering layers." *Nature* 491.7423 (2012): 232-234.

[2] Boniface, Antoine, Jonathan Dong, and Sylvain Gigan. "Non-invasive focusing and imaging in scattering media with a fluorescence-based transmission matrix." *Nature communications* 11.1 (2020): 1-7.

[3] Ruan, Haowen, et al. "Deep tissue optical focusing and optogenetic modulation with time-reversed ultrasonically encoded light." *Science advances* 3.12 (2017): eaao5520.

Reply:

Thanks for your thoughtful discussions and the suggestion.

We have removed the non-invasive claim from the title and abstract of the paper. In addition, the discussion of the potential strategies toward non-invasive approach was added in Discussion part in the main article.

We changed the title to "Robust and adjustable dynamic scattering compensation for high-precision deep tissue optogenetics". Also, in line 18, we changed the original text to "high-precision optogenetics".

In line 347, the "noninvasive" was also replaced by "high-precision". From line 352 to line 357, we moved the end part of next paragraph to here and discuss the corresponding potential strategies toward noninvasive approach as follows:

"Using the wavelengths mentioned above, our system increases the penetration depth up to 500 μm . Considering the penetration depth of long wavelength (infrared) light is much longer than the visible light, the system can increase its working distance to a brand-new

level. Furthermore, using long wavelength multiphoton-excited fluorescence as the feedback signal, a reflection setup can be implemented to noninvasive in vivo experiments.”

We understand your concern about the not fully noninvasive system design. Previously, we regard invasive point source as the basis for judging whether the system is noninvasive, as in some other articles^{3,4}, where the light source and the camera were also not on the same side. Our main purpose of this is to separate this kind of focusing strategy from the fiber-based approaches-obtaining a tight focus by modulating light field outside the tissue quickly and precisely rather than using optical fibers to invasively guide light to target locations. Regarding that the spatial resolution of fiber approaches is also low, we consider that changing the word “noninvasive” to “high-precision” can also separate these two kinds of strategies properly since the key point here is to form a focus after scattering because just noninvasive but low in spatial resolution does not make much sense.

A major obstacle for the strategy to perform in reflection mode is the collection of signal light. Using the nonlinear relationship between fluorescence intensity and excitation light intensity in multiphoton mode, the fluorescence light can be gathered as signal, and the reflection operating mode can be achieved to promote the development of fully noninvasive approaches. Furthermore, most invasive methods require the implantation position to be determined in advance while our proposed strategy can move the target flexibly. Such ability to direct the light to the target outside the tissue is necessary to perform fully noninvasive experiments and we believe our strategy is promising to achieve this purpose.

1. Zhang Z, Russell LE, Packer AM, Gauld OM, Hausser M. Closed-loop all-optical interrogation of neural circuits in vivo. *Nat Methods* **15**, 1037-1040 (2018).
2. Accanto N, *et al.* A flexible two-photon fiberscope for fast activity imaging and precise optogenetic photostimulation of neurons in freely moving mice. *Neuron*, (2022).
3. Wu T, Dong J, Gigan S. Non-invasive single-shot recovery of a point-spread function of a memory effect based scattering imaging system. *Opt Lett* **45**, 5397-5400 (2020).
4. Wang D, Sahoo SK, Zhu X, Adamo G, Dang C. Non-invasive super-resolution imaging through dynamic scattering media. *Nat Commun* **12**, 3150 (2021).